# Microbial tryptophan metabolism activates host lysosomal activity to facilitate lipid breakdown

**Kenan Zhang**[☉], **Zihan Luo**[☉], **Yan Chen, Yan Li, Lang Wang, Yanan Liu, Ruizhi Yang, Qian Li, Jiahao Zhao, Bin Qi** *, **Zhao Shan** *

School of Life Sciences, Center for Life Sciences, Southwest United Graduate School, Yunnan Key Laboratory of Cell Metabolism and Diseases, Yunnan University, Kunming, China

[☉] These authors contributed equally to this work.
* qb@ynu.edu.cn (BQ); shanzhaolab@163.com (ZS)

## Abstract

Lysosomes are central to lipid metabolism, yet how gut microbiota-derived metabolites regulate lysosomal function to influence host lipid homeostasis remains unknown. Here, we identify a mechanism in which bacterial tryptophan metabolism activates lysosomal activity to promote lipid breakdown in *Caenorhabditis elegans*, and show that the bacterial tryptophan metabolite indole recapitulates these effects in mammalian hepatocytes. By developing a lysosomal-responsive lipid reporter in *C. elegans* to screen for bacterial metabolic states that modulate host lipid storage, we discover that *Escherichia coli* tryptophan catabolism via tryptophanase TnaA induces lysosomal lipid chaperone LBP-8, driving lipid mobilization. Moreover, tryptophan metabolite indole enhanced lysosomal acidification and degradation capacity, while genetic disruption of lysosomal regulators reversed these effects. Strikingly, bacterial tryptophan metabolism further promoted mitochondrial β-oxidation through lysosomal lipase activity. This pathway was conserved in mammalian hepatocytes, where *E. coli*-derived tryptophan metabolite indole enhances lysosomal function and reduce lipid accumulation. Our work uncovers microbiota-regulated lysosomal activation as a critical axis in lipid homeostasis, highlighting its potential as a therapeutic target for metabolic disorders linked to lysosomal dysfunction.

## Introduction

The gut microbiota plays a pivotal role in regulating host metabolism, including lipid homeostasis, through the production of bioactive metabolites that influence cellular processes such as energy storage, mitochondrial function, and organelle dynamics [1,2]. Among these processes, lysosomes serve as central hubs for lipid degradation, where acid hydrolases break down complex lipids into free fatty acids for energy production or recycling [3]. Mitochondria, essential for energy production and evolutionarily derived from bacteria [4], communicate with bacteria to regulate host immune responses [5], metabolism [6,7] and aging [8]. While the impact of bacterial

**Data availability statement:** All relevant data is available within the manuscript and Supporting information files. All numerical data are found in S1 Data, which contains multiple datasheets. Sequencing data of RNA-seq have been deposited in CNCB under accession codes PRJCA056633 (https://ngdc. cncb.ac.cn/bioproject/browse/PRJCA056633) and PRJCA056670 (https://ngdc.cncb.ac.cn/bioproject/browse/PRJCA056670).

**Funding:** This work was supported by the Yunnan Provincial Science and Technology Project at Southwest United Graduate School (https://kjt.yn.gov.cn/,202302AP370005 to B.Q.), National Natural Science Foundation of China (https://www.nsfc.gov.cn/, 82570734 to Z.S.; 32541028 to B.Q.), Yunnan Revitalization Talent Support Program (https://ylxf.1237125.cn/, C619300A086 to Z.S., K264202230211 to B.Q.). The funders had no role in study design, data collection and analysis, decision to publish, or preparation of the manuscript.

**Competing interests:** I have read the journal's policy and the authors of this manuscript have the following competing interests: Z.S., B.Q., K.Z., and Z.L. are inventors on Chinese patent application-"The use of Escherichia coli in combination with tryptophan for lysosomal function activation" (application no. 202510593904.8, filed May 09, 2025). All other authors have no competing interests.

**Abbreviations:** ACDH, acyl-CoA dehydrogenases; ACS, acyl-CoA synthetases; BafA1, Bafilomycin A1; BMP, bis(monoacylglycero) phosphate; CEs, cholesterol esters; CGC, *Caenorhabditis* Genetics Center; CTSD, cathepsin D; DMSO, dimethyl sulfoxide; ER, endoplasmic reticulum; FBS, fetal bovine serum; FFAs, free fatty acids; GO, Gene Ontology; IAA, indole-3-acetic acid; ILVs, intra-luminal vesicles; LAL, lysosomal acid lipase; LTR, LysoTracker Red; NAFLD, non-alcoholic fatty liver disease; NGM, nematode growth medium; OA, oleic acid; ORO, Oil Red O; PA, palmitic acid; PFA, paraformaldehyde; ROS, reactive oxygen species; TGs, triglycerides; tBuOOH, tert-butyl hydroperoxide.

metabolites on mitochondria is becoming increasingly recognized in recent years [2], the specific regulatory roles of these metabolites on lysosomal function remain largely unexplored.

Lysosomes are central to lipid metabolism, acting as enzymatic hubs for fat breakdown and recycling. In mammals, lysosomal acid lipase (LAL) is indispensable, degrading triglycerides (TGs), cholesterol esters (CEs), and lipoproteins in the acidic lysosomal environment [3]. In *Caenorhabditis elegans,* major fats are stored in vesicles distinct from lysosome-related organelles [9]. Lysosomal lipases are essential for lipid droplet (LD) degradation, as their absence results in LD accumulation [10]. Beyond breakdown, these lipases regulate lipid signaling: LIPL-4 activation triggers a lysosome-to-nucleus cascade that boosts lipolysis and extends life span [11]. This pathway involves LIPL-4 and its lipid chaperone LBP-8, which mobilizes fatty acids to enhance mitochondrial β-oxidation, reduce lipid storage, and promote longevity [12]. Dysregulation of lysosomal function disrupts lipid homeostasis and is implicated in pathologies ranging from lysosomal storage disorders [13] to metabolic diseases [14]. Despite their central role in lipid handling, the mechanisms by which extrinsic factors—such as gut microbial metabolites—modulate lysosomal activity remain poorly understood. Identifying bacterial metabolites that fine-tune lysosomal function is thus essential for unraveling host-microbe metabolic crosstalk and developing therapies for lysosomal-related disorders.

Gut bacterial metabolism of tryptophan plays a pivotal role in animal physiology, generating indole derivatives and other metabolites that regulate immune responses, intestinal barrier integrity, and systemic metabolism [15–18]. In metabolic disorders such as obesity, type 2 diabetes, and non-alcoholic fatty liver disease (NAFLD), compositional and functional alterations in the gut microbiota are well-documented [19], with impaired bacterial tryptophan metabolism emerging as a key contributor to disease pathogenesis [20]. However, how microbial tryptophan metabolism regulates cellular organelle dynamics in host is still unclear. Given the central role of lysosomes in lipid catabolism and their dysregulation in metabolic diseases, understanding whether gut-derived tryptophan metabolites modulate lysosomal activity could unveil novel therapeutic targets. Addressing this gap is critical for leveraging host–microbe crosstalk to restore lipid metabolism and combat lysosomal-related disorders.

In this study, we developed a lysosomal-responsive lipid reporter system in *C. elegans* to screen for bacterial metabolic states that modulate host lipid storage. Our findings reveal an evolutionarily conserved mechanism in which bacterial tryptophan metabolite, indole, activates lysosomal activity to drive lipid catabolism—a process validated in both *C. elegans* and mammalian models. These results deepen our understanding of inter-organismal metabolic crosstalk and establish a framework for therapies targeting microbial tryptophan metabolic pathways to regulate lysosomal function and counteract lipid-related pathologies.

## Results

### Establishment of a lysosomal-responsive lipid reporter system to screen bacterial metabolic state changes regulating lipid metabolism

The constitutive activation of lysosomal lipolysis via overexpression of lysosomal acid lipase LIPL-4 drives nuclear trans-location of signaling molecules, including the lipid chaperone LBP-8 and the lipid messenger oleoylethanolamine, which transcriptionally activate metabolic genes to promote fat mobilization [11]. Specifically, the LIPL-4–LBP-8 lysosomal signaling pathway enhances mitochondrial β-oxidation, leading to reduced total fat storage in animals overexpressing *lipl-4* or its downstream effector *lbp-8* [12]. This suggests an inverse correlation between *lbp-8* expression levels and lipid accumulation, where elevated *lbp-8* expression correlates with reduced lipid stores (Fig 1A). To validate this relationship, we generated transgenic *C. elegans* overexpressing *lbp-8* and quantified lipid levels using Oil Red O (ORO) staining. As predicted, *lbp-8*-overexpressing animals (*lbp-8 Tg*) exhibited significantly reduced lipid accumulation compared to wild-type controls (Fig 1B).

To determine whether bacterial metabolic states influence *lbp-8* expression and subsequent lipid metabolism, we employed an reporter which involved in LIPL-4–LBP-8 lysosomal signaling pathway for lipid metabolism [12], P*lbp-8*::GFP transcriptional reporter (Fig 1C). We first tested this by culturing *C. elegans* under standard conditions on NGM plates seeded with metabolically compromised *Escherichia coli* (either antibiotic-treated with ampicillin or UV-killed). However, no significant changes in P*lbp-8*::GFP expression were observed under these conditions (S1A Fig).

Remarkably, when we cultured worms on nutrient-rich LB medium plates seeded with *E. coli*-K12, we observed significant induction of *lbp-8* at both the transcriptional (Fig 1D) and translational (Fig 1E) levels. The LB medium condition promoted robust bacterial overgrowth, resulting in thick bacterial lawns (S1B Fig), suggesting that the nutrient-rich environment may alter *E. coli* metabolism that induces *lbp-8* expression. To determine whether LB medium components affect *lbp-8* expression independently of bacterial presence, we examined P*lbp-8*::*lbp-8*::*GFP* reporter expression in worms exposed to bacteria-free NGM or LB plates (S1C Fig). The results demonstrate that LB medium alone induces a slight increase in *lbp-8* expression compared to NGM (S1C Fig), likely attributable to its higher amino acid content, including tryptophan. However, this effect is substantially weaker than the induction observed in worms exposed to bacteria-cultured LB plates compared to bacteria-cultured NGM plates (S1C Fig). This finding suggests that while medium components contribute minimally, bacterial metabolism is the primary driver of *lbp-8* induction. To distinguish whether the effect of *lbp-8* expression is driven by bacterial viability, metabolic activity, we compared *lbp-8* expression in worms fed with live *E. coli* K12, heat-killed *E. coli* K12 (see Methods), or no bacteria, on both NGM and LB plates (S1C Fig). Live bacteria cultured in LB medium induced *lbp-8* expression more strongly than heat-killed bacteria (S1C Fig), indicating that active metabolism or bacterial viability during feeding is responsible for *lbp-8* induction.

Consistent with the role of LBP-8 in lysosomal signaling, nuclear translocation of LBP-8—a hallmark of lysosomal lipolysis activation [11]—was significantly enhanced in animals fed *E. coli* on LB plates (Fig 1F). Correspondingly, ORO staining revealed significantly reduced lipid stores in these animals (Fig 1G). Together, these findings suggested that bacterial metabolic changes induced by LB culture conditions can trigger LBP-8 expression and promote lipid breakdown in *C. elegans*.

### *E. coli* tryptophan metabolism induces *lbp-8* expression and subsequent lipid breakdown

To identify bacterial metabolic factors modulating *lbp-8* activation and lipid mobilization in *C. elegans*, we conducted a genome-wide screen of the *E. coli* single-gene knockout library (Keio collection). Using the parental strain *E. coli* K-12 BW25113 as a control, we assessed P*lbp-8*::GFP expression under LB-medium plate culture (Fig 2A). After screening an *E. coli* single-gene knockout library, we found that 19 *E. coli* mutant significantly reduced *lbp-8* expression (Figs 2A, S2A, and S2B). GO analysis of the 19 screened *E. coli* mutants revealed that the "Metabolic Process" category contains the

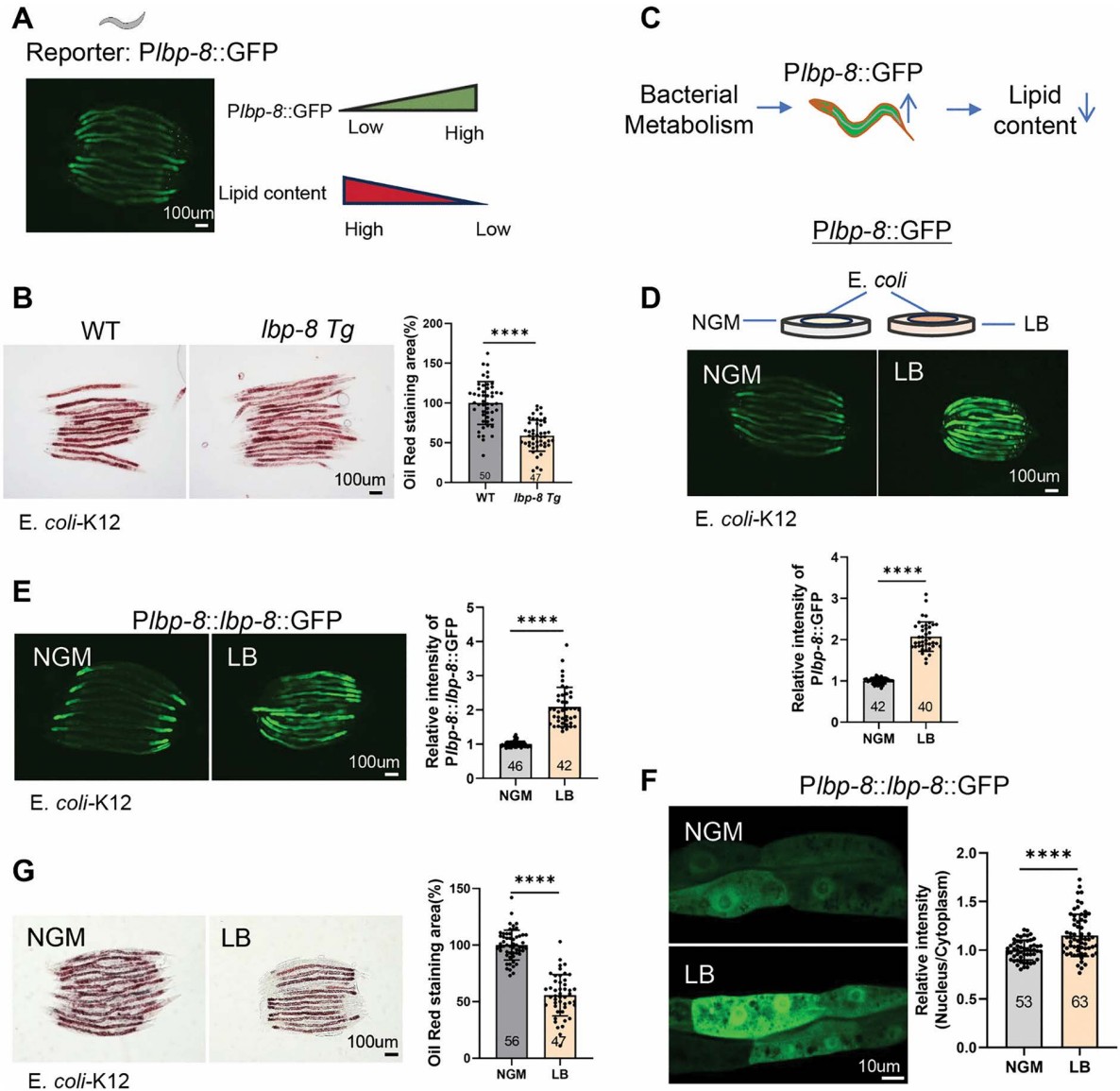

**Fig 1. Establishment of a lysosomal-responsive lipid reporter in *C. elegans* to screen bacterial metabolic state changes regulating lipid metabolism. (A)** Schematic of the inverse relationship between *lbp-8* expression and fat accumulation, alongside representative fluorescence micrographs of the P*lbp-8*::GFP reporter in wild-type L4 larvae. Scale bar, 100 μm. **(B)** Oil Red O staining and quantification of total lipid levels in wild-type (WT) vs. *lbp-8*–overexpressing animals (*lbp-8 Tg*) at the L4 stage. The number of animals analyzed is indicated. Scale bar, 100 μm. **(C)** Schematic model illustrating that changes in the bacterial metabolic state induce *lbp-8* expression, thereby promoting lipid breakdown. **(D)** Fluorescence micrographs and quantification of P*lbp-8*::GFP reporter expression in L4 larvae fed on standard NGM plates seeded with *E. coli* K12 (NGM) vs. LB-conditioned *E. coli* K12 (LB). The number of animals analyzed is indicated. Scale bar, 100 μm. **(E)** Fluorescence images and quantification of the P*lbp-8*::*lbp-8*::GFP expression in L4 larvae under the same dietary conditions as in **(D)**. The number of animals analyzed is indicated. Scale bar, 100 μm. **(F)** High-magnification fluorescence images and quantification of the nuclear-to-cytoplasmic GFP intensity ratio for LBP-8::GFP in L4 larvae on NGM vs. LB *E. coli*. The number of animals analyzed is indicated. Scale bar, 10 μm. **(G)** Oil Red O staining and quantification of lipid content in wild-type L4 larvae fed on NGM vs. LB *E. coli*-K12. The number of animals analyzed is indicated.Scale bar, 100 μm. Data represent mean ± SD. All statistical analyses were performed using unpaired two-tailed Student *t* test. ****$p < 0.0001$. All experiments were performed independently at least three times. The data underlying this Figure can be found in S1 Data.

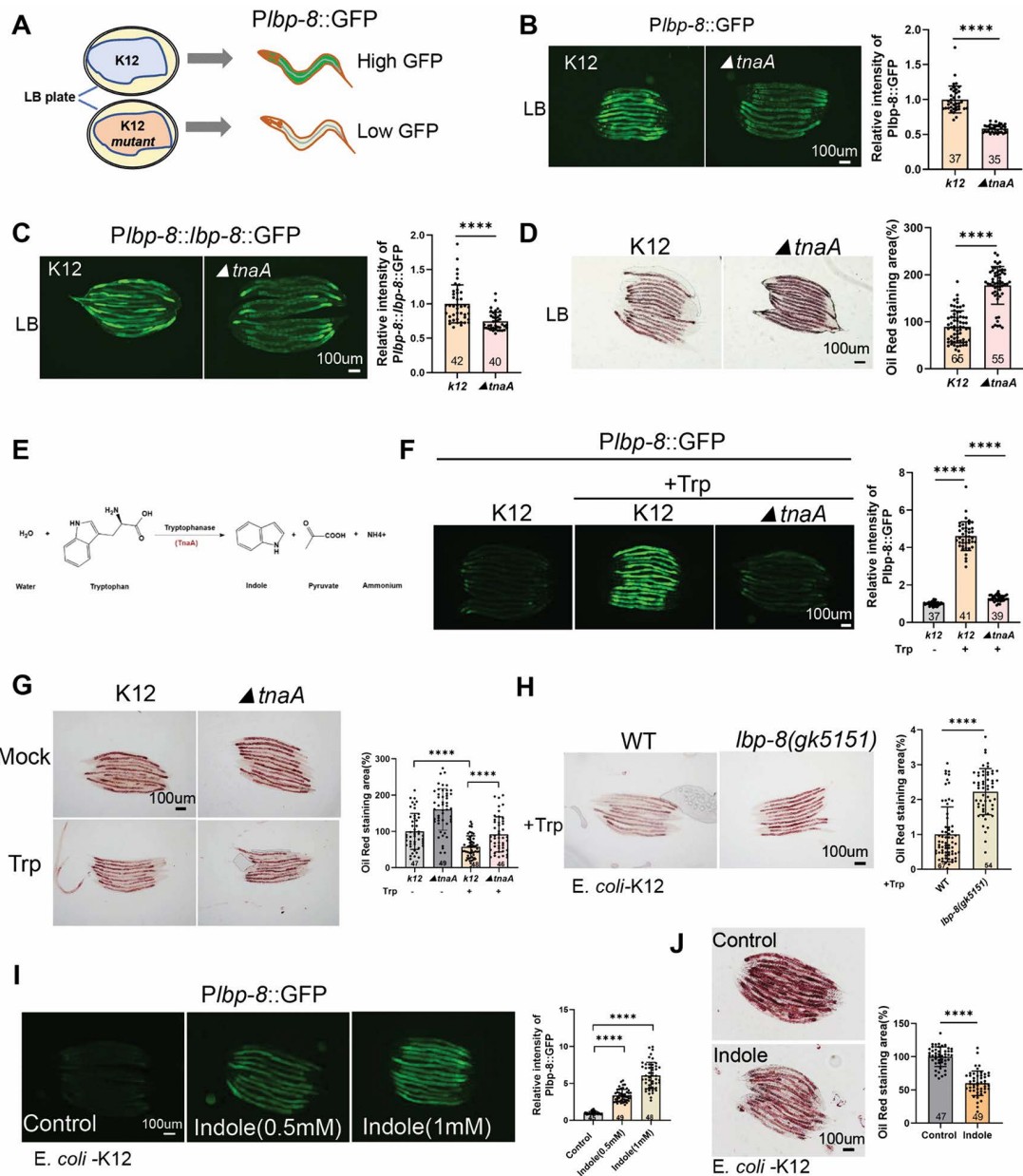

**Fig 2. Bacterial tryptophan metabolism regulates lipid homeostasis. (A)** Schematic of a genome-wide screen using the *E. coli* single-gene knockout library in *C. elegans* harboring the P*lbp-8*::GFP reporter, identifying *E. coli* mutants that fail to induce GFP on LB medium. **(B)** Representative fluorescence images and quantification of the P*lbp-8*::GFP reporter in L4 animals feed on LB medium seeded with wild-type *E. coli* K12 or the Δ*tnaA* mutant. The number of animals analyzed is indicated. Scale bar, 100 μm. **(C)** Fluorescence images and quantification of the P*lbp-8*::*lbp-8*::GFP reporter under the same conditions as in **(B)**. The number of animals analyzed is indicated. Scale bar, 100 μm. **(D)** Oil Red O staining and quantification of lipid content in wild-type L4 animals feed on LB medium with wild-type *E. coli* K12 or Δ*tnaA*. The number of animals analyzed is indicated. Scale bar, 100 μm. **(E)** Diagram of the tryptophan catabolic pathway in *E. coli*, highlighting the role of TnaA. **(F)** Fluorescence images and quantification of the P*lbp-8*::GFP reporter in L4 animals feed on NGM plate seeded with *E. coli*- K12, or 10 mM tryptophan-supplemented NGM plated seeded with *E. coli*- K12 or Δ*tnaA*. The number of animals analyzed is indicated. Scale bar, 100 μm. **(G)** Oil Red O staining and quantification of lipid levels in wild-type L4 larvae fed on NGM plate with or without 10 mM tryptophan, seeded with K12 or Δ*tnaA*. The number of animals analyzed is indicated. Scale bar, 100 μm. **(H)** Oil Red O staining and quantification of wild-type and *lbp-8*(gk5151) mutant L4 worms fed on NGM plate supplemented with 10 mM tryptophan and seeded with K12. The number of animals analyzed is indicated. Scale bar, 100 μm. **(I)** Fluorescence images and quantification of the P*lbp-8*::GFP reporter in L4 animals fed on NGM plate with or without indole, seeded with *E. coli* K12. The number of animals analyzed is indicated. Scale bar, 100 μm. **(J)** Oil Red O staining and quantification of lipid levels in wild-type L4 larvae fed on NGM plate with or without 0.5 mM indole, seeded with *E. coli* K12. The number of animals analyzed is indicated. Scale bar, 100 μm. Data represent mean±SD. All statistical analyses were performed using unpaired two-tailed Student *t* test. ****$p < 0.0001$. All experiments were performed independently at least three times. The data underlying this Figure can be found in S1 Data.

most genes (tnaA, astE, hcaB, glpK, glgA, ubiH) (S2C Fig), indicating that bacterial metabolic activity is the primary factor affecting *lbp-8* expression. Since LB plates are rich in tryptone—a source of amino acids including tryptophan—this raised the possibility that bacterial tryptophan metabolism modulates host lipid breakdown by inducing *lbp-8* expression. Notably, among all the screened candidates, *tnaA*(tryptophanase) is the only gene directly involved in tryptophan metabolism. Therefore, we focused on investigating how *E. coli* TnaA-dependent metabolism promotes *lbp-8* expression.

The *E. col*-tnaA mutant exhibited the robust effect on *lbp-8* expression. Deletion of tnaA markedly suppressed *lbp-8* expression at both transcriptional (Fig 2B) and translational (Fig 2C) levels under LB-medium plate conditions. Transcriptomic analysis confirmed that tnaA deficiency abolished LB-induced *lbp-8* upregulation (S3A Fig), while nuclear translocation of LBP-8 was similarly reduced in animals fed *E. coli*-tnaA (S3B Fig). Notably, ORO staining revealed elevated total lipid levels in *C. elegans* fed tnaA mutants on LB plates (Fig 2D), indicating that *E. coli* TnaA-dependent metabolism promotes *lbp-8* expression and subsequent lipid breakdown.

### Indole, a bacterial tryptophan metabolite, regulates LBP-8 levels and lipid storage

TnaA, a tryptophanase, catalyzes tryptophan conversion into indole, pyruvate, and ammonium (Fig 2E), which raised the possibility that *E. coli* tryptophan metabolism modulates host lipid breakdown by inducing *lbp-8* expression. To test this, we supplemented standard NGM plates (seeded with wild-type *E. coli*) with tryptophan, which robustly induced *lbp-8* expression (Figs 2F, S3C, and S3E) and LBP-8 nuclear translocation (S3D Fig). In contrast, supplementing tryptophan into plates with heat-killed *E. coli* did not induce *lbp-8* expression (S3F Fig), underscoring the necessity of active bacterial metabolism for *lbp-8* regulation.

Furthermore, while tryptophan supplementation induced *lbp-8* expression in animals fed wild-type *E. coli*, this induction was lost when worms were fed the *E. coli* -tnaA mutant (Fig 2F). Correspondingly, lipid levels decreased in wild-type-fed worms with tryptophan supplementation (Fig 2G) but were restored in animals fed the tnaA mutant. Finally, *lbp-8* mutant animals exhibited higher lipid levels despite tryptophan supplementation when fed wild-type *E. coli* (Fig 2H), suggesting that bacterial tryptophan metabolism promotes lipid breakdown via LBP-8-dependent pathway. In summary, our results demonstrate that bacterial TnaA-mediated tryptophan metabolism induces *lbp-8* expression, which in turn drives lipid breakdown in *C. elegans*.

To determine which specific metabolite of TnaA-catalyzed tryptophan catabolism upregulates LBP-8 levels, we supplemented NGM plates with each of the three downstream products—indole, pyruvate, ammonium chloride, and ammonium sulfate—and examined *lbp-8* expression. Our results revealed that indole supplementation specifically and significantly induced *lbp-8* expression and nuclear translocation of LBP-8 (Figs 2I and S3G), while pyruvate and ammonium did not show similar effects. Furthermore, indole supplementation reduced lipid stores, as demonstrated by ORO staining (Fig 2J) and decreased lipid droplets observed using the DHS-3::GFP reporter (S3I Fig). These findings identify indole as a specific secondary metabolite derived from bacterial tryptophan metabolism that regulates host LBP-8 levels and lipid storage in *C. elegans*.

### *E. coli* tryptophan metabolism induce lysosomal-related genes expression

To further elucidate the molecular mechanisms by which bacterial tryptophan metabolism modulates host lipid metabolism, we performed RNA-sequencing (RNA-seq) on *C. elegans* under two experimental conditions: (1) wild-type animals fed *E. coli* K12 (Mock-K12) versus those fed K12 supplemented with tryptophan (Trp-K12), and (2) Trp-K12 versus animals fed tnaA mutants in tryptophan-supplemented medium (Trp-tnaA) (S4A Fig). The transcriptomic analysis revealed that tryptophan supplementation in the K12 group upregulated approximately 2,050 genes while downregulating 1,285 genes compared to the mock treatment. In contrast, comparing Trp-K12 to Trp-tnaA animals showed 1648 genes upregulated and 1,437 downregulated (S4B Fig and S1 Table), indicating that the presence of bacterial TnaA is essential for a full transcriptional response to tryptophan-derived metabolites.

Focusing on lipid metabolism, we observed that key lipases responsible for the breakdown of triacylglycerol into free fatty acids—including adipose TG lipase (ATGL/*atgl-1*) and hormone-sensitive lipase (HSL/*hosl-1*)—as well as LALs, were significantly induced by tryptophan metabolism (Fig 3A). Additionally, the lipid-binding proteins LBP-1 and LBP-8 were specifically induced, further linking bacterial tryptophan metabolism to enhanced lipid mobilization. In contrast, genes encoding enzymes involved in lipid synthesis, such as POD-2 (acetyl-CoA carboxylase), FASN-1 (fatty acid synthase), several elongases (*elo-2*, *elo-8*), and desaturases (*fat-6*, *fat-7*), remained unchanged or were suppressed (Fig 3A). This result suggested that bacterial tryptophan metabolic mainly induce lysosomal lipases expression which may facilitate breakdown of triacylglycerol into free fatty acids.

KEGG pathway analysis of differentially expressed genes (Trp-K12 versus Mock-K12) underscored the lysosome as a key pathway responsive to bacterial tryptophan metabolites (S4C Fig), with 14 lysosome-related genes (S4D Fig)—including various lipohydrolases and other hydrolases—being significantly induced at both group (Trp-K12 versus Mock-K12, Trp-K12 versus Trp-tnaA)(S4E and S4F Fig). Given that lysosomes play a critical role in maintaining lipid metabolism by breaking down and recycling various lipid species. They contain a variety of acid hydrolases—including LAL—that digest lipids such as TGs, CEs, and other complex lipids into free fatty acids and cholesterol [3]. Thus, it is possible that bacterial tryptophan metabolism regulates lysosomal function to mediates lipid metabolism in *C. elegans*.

### *E. coli* tryptophan metabolism activates lysosomal function

To investigate whether bacterial tryptophan metabolism modulates lysosomal activity, we systematically evaluated lysosomal morphology, acidification, and degradation capacity in *C. elegans* fed *E. coli* K12 or tnaA mutants under tryptophan-supplemented conditions.

First, we tracked age-associated lysosomal morphology changes using the established reporter (XW5399:*qxIs257* [$P_{ced-1}$NUC-1::CHERRY]) [21,22] fed K12 (Mock-K12), K12 with tryptophan (Trp-K12), or tnaA mutants with tryptophan (Trp-tnaA). Consistent with prior reports [22], Mock-K12 animals (grown on K12-*E. coli* without tryptophan) exhibited age-dependent lysosomal remodeling, including reduced vesicular structures and increased tubular morphology. In contrast, Trp-K12 animals (fed K12-*E. coli* with tryptophan) maintained a shorter tubular lysosomes and higher proportion of vesicular lysosomes through aging (at Day 3 and Day 6), but not at Day 1 (S5A Fig). Strikingly, Trp-tnaA animals (fed tnaA mutant *E. coli* with tryptophan) mirrored Mock-K12 morphology, by increasing tubular lysosomes and reducing vesicular structures with aging (S5A Fig). This data indicates that bacterial tryptophan metabolism preserves a youthful lysosomal architecture.

Next, we assessed lysosomal acidification using the transgenic reporter NUC-1::pHTomato (XW19180: *qxIs750*[$P_{hs}$NUC-1::pHTomato]) [22]. In this system, the pH-sensitive fluorescent protein pHTomato (pKa ≈ 7.8) [23] is fused to NUC-1 and expressed under a heat-shock promoter; pHTomato decreases in fluorescence at lower pH values (Fig 3B). We found that Trp-K12 animals (fed K12-*E. coli* with tryptophan) exhibited significantly lower NUC-1::pHTomato fluorescence compared to Mock-K12 animals (grown on K12-*E. coli* without tryptophan) (Fig 3C). Importantly, this reduction in fluorescence was reversed in Trp-tnaA animals (fed tnaA mutant *E. coli* with tryptophan) (Fig 3C), indicating that bacterial tryptophan metabolism enhances lysosomal acidification—a condition that is critical for the optimal activity of lysosomal hydrolases.

Finally, we evaluated lysosomal degradation activity using the transgenic reporter- *qxIs257*, $P_{ced-1}$::NUC-1::CHERRY [21]. When NUC-1::CHERRY delivered to lysosomes, CHERRY is cleaved by cathepsins, and the cleaved products can be quantified by western blot (Fig 3D). Our results showed that, following tryptophan supplementation in *E. coli* K12, there was a induction in cleaved CHERRY levels; in contrast, no such induction was observed in animals fed the tnaA mutant (Fig 3E). Additionally, using a cathepsin assay with MagicRed [24]—a substrate that emits red fluorescence upon cleavage by Cathepsin B—we observed increased lysosomal degradation activity in animals fed *E. coli* HT115 with tryptophan supplementation (Fig 3F). This data indicates that bacterial tryptophan metabolism increases lysosomal degradation activity in *C. elegans*.

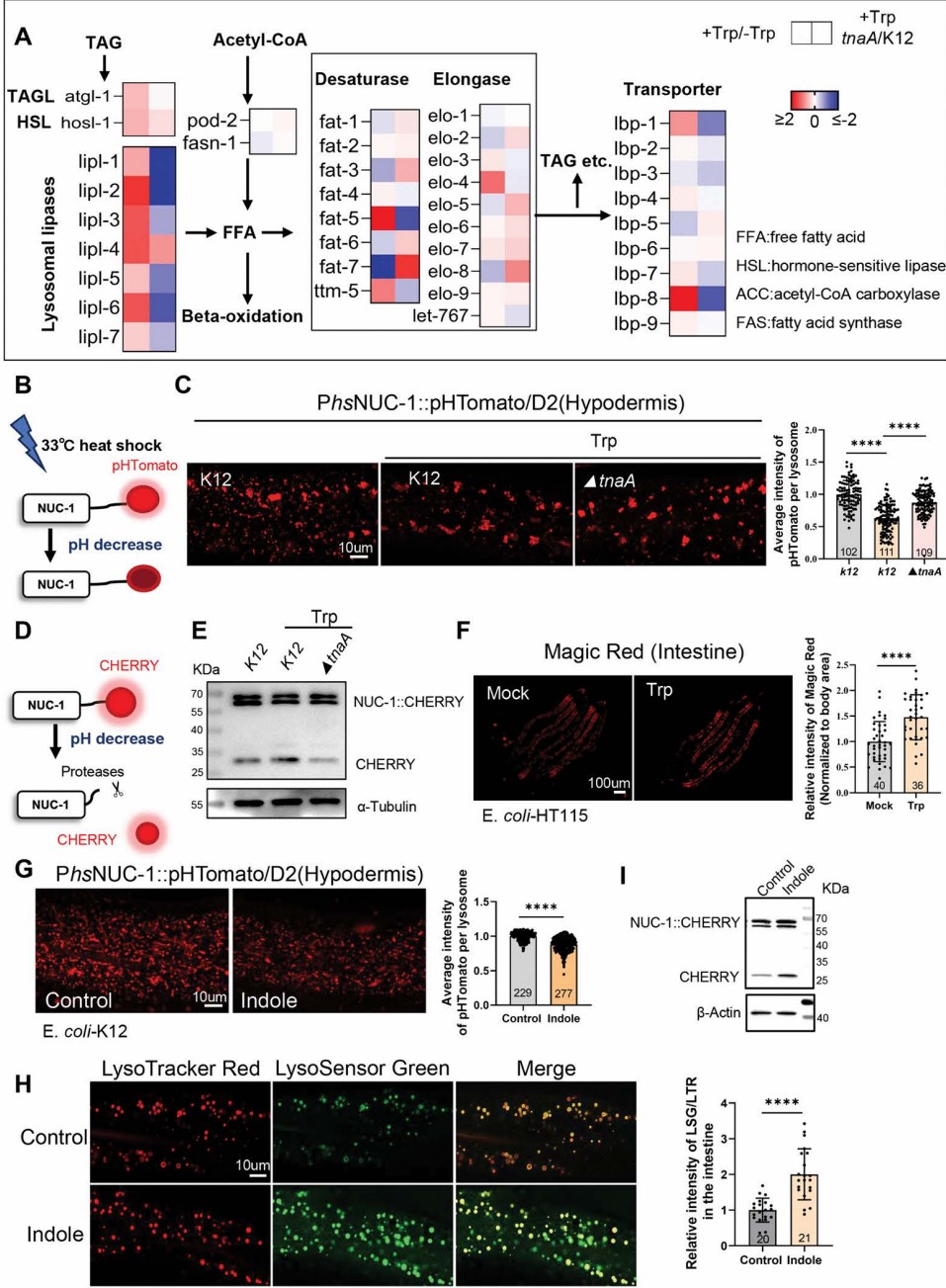

**Fig 3. *E. coli* tryptophan metabolism activates lysosomal function. (A)** Heat map showing fold changes in mRNA levels of lipid-metabolism genes in worms fed Trp-supplemented K12 versus mock-treated K12, and Trp-supplemented ΔtnaA versus Trp-supplemented K-12. Fold changes were calculated by dividing each gene's expression level under the indicated conditions. Source data are provided in S1 Table. **(B)** Schematic of the heat-shock-inducible NUC-1::pHTomato reporter. Fusion of the lysosomal nuclease NUC-1 to the pH-sensitive fluorophore pHTomato allows real-time monitoring of lysosomal pH. Fluorescence decreases as lysosomal acidification increases (i.e., as pH declines). **(C)** Confocal micrographs and quantitative analysis of lysosomes in day-2 adults expressing hsNUC-1::pHTomato after heat shock. Animals were maintained on NGM plates seeded with K12 (control) or with K12/ΔtnaA plus 10 mM tryptophan. The average intensity of pHTomato per lysosome is shown in the right. The number of lysosomes (indicated in the Figure) from at least 20 animals were scored in each group. Scale bar, 10 μm. **(D)** Diagram of the lysosomal degradation assay based on cleavage of NUC-1::mCherry in worms. **(E)** western blot of mCherry cleavage products from NUC-1::mCherry in day-1 adults fed K12 or K-12/ΔtnaA with tryptophan, demonstrating enhanced lysosomal degradation. **(F)** Representative fluorescence images (left) and quantification (right) of Magic Red cathepsin activity in L4 worms fed on NGM plate with or without 10 mM tryptophan, seeded with *E. coli*-HT115. The number of animals analyzed is indicated. Scale bar,

100 μm. **(G)** Confocal micrographs (left) and quantitative analysis (right) of lysosomes in day-2 adults expressing hsNUC-1::pHTomato after heat shock. Animals were maintained on *E. coli*-K12 seeded NGM plated with or without 0.5mM indole supplementation. The average intensity of pHTomato per lysosome is shown in the right. The number of lysosomes (indicated in the Figure) from at least 20 animals were scored in each group. Scale bar, 10 μm. **(H)** Confocal fluorescence images (left) and quantification (right) of the intestine day-2 adults wild-type animals stained by LysoTracker Red (LTR DND-99) and LysoSensor Green (LSG DND-189). Animals were maintained on *E. coli*-K12 seeded NGM plated with or without 0.5mM indole supplementation. The relative intensity of LSG/LTR in wild type is shown in the right. Scale bar,100 μm. **(I)** western blot of mCherry cleavage products from NUC-1::mCherry in day-1 adults maintained on *E. coli*-K12 seeded NGM plated with or without 0.5mM indole supplementation. Data represent mean±SD. All statistical analyses were performed using unpaired two-tailed Student *t* test. ****$p<0.0001$. All experiments were performed independently at least three times. The data underlying this Figure can be found in S1 Data.

Furthermore, we verified the effect of the tryptophan secondary metabolite indole on lysosomal activity. In the hypodermis, indole supplementation reduces the average fluorescence of NUC-1::pHTomato in epidermal lysosome (Fig 3G), indicating increased lysosomal acidification. To assess intestinal lysosomal acidity in the intestine, we performed co-staining with LysoTracker Red (LTR) and LysoSensor Green DND-189 (LSG, pKa 5.2) [25]. Since LTR is less sensitive to pH changes than LSG, it serves as a normalization control for dye uptake [26]. The LSG/LTR fluorescence intensity ratio was quantified as an indicator of lysosomal acidity. We found that indole supplementation significantly increases the LSG/LTR ratio (Fig 3H) and intensity of LysoSensor Green (S5B Fig) in wild-type animals at day 2, confirming enhanced intestinal lysosomal acidification. Additionally, we examined CHERRY cleavage from the fusion protein NUC-1::CHERRY by cathepsin in lysosomes (Fig 3I), demonstrating that lysosomal proteolytic activity is also upregulated by indole. Taken together, these data show that the acidity of both epidermal and intestinal lysosomes, as well as lysosomal degradation activity, are all enhanced by indole. These findings suggest that the bacterial tryptophan metabolite indole promotes host lysosomal function across multiple tissues.

### *E. coli* tryptophan metabolism drives lysosomal-dependent lipid degradation

To test whether bacterial tryptophan metabolism reduces host lipid content by enhancing lysosomal activity, we genetically disrupted lysosomal function and assessed its impact on *lbp-8* expression and lipid levels.

First, we targeted *cup-5*, encoding a lysosomal Ca²⁺ channel homologous to human TRPML that regulates lysosomal acidity and activity [21,27,28]. Knockdown of *cup-5* in Trp-K12 animals (fed tryptophan-supplemented *E. coli* K12) abolished the tryptophan-induced reduction in lipid content (Fig 4A) and suppressed *lbp-8* upregulation (S6A and S6B Fig). Similarly, knockdown of *lipl-4*—a LAL required for lipid hydrolysis at low pH [11]—restored lipid levels (Fig 4A) and attenuated *lbp-8* induction in Trp-K12 animals (S6A and S6B Fig). These findings demonstrate that bacterial tryptophan metabolism drives lipid degradation through lysosomal acidification and lipase activation.

Lysosomal lipid degradation—whether of lipoproteins, lipid droplets, or cell membranes—occurs predominantly within the lysosomal lumen and depends on the formation of intraluminal vesicles (ILVs). A specific phospholipid, bis(monoacylglycero)phosphate (BMP), is critical for ILV formation and BMP-mediated lipid degradation [29,30]. Previous studies have reported that lysosomal enzymes PLD3 and PLD4 are necessary for maintaining normal BMP levels in human cells and murine tissues [30], thereby regulating lipid degradation in lysosomes. We found that knockdown of F09G2.8 (homologous to PLD3/PLD4) or *pld-1* (a phospholipase D homolog) (S6C Fig) partially reversed the activation of *lbp-8* and the reduction in lipid content in Trp-K12 animals (Fig 4B and 4C). This demonstrates that bacterial tryptophan metabolism relies on BMP-dependent ILV formation to promote lysosomal lipid degradation. Collectively, these findings indicate that bacterial tryptophan metabolism promotes lipid degradation in *C. elegans* through functional lysosomes.

### *E. coli* tryptophan metabolism enhances mitochondrial β-oxidation through lysosomal activation to promote lipid metabolism

To identify regulatory factors driving increased lipid mobilization in *C. elegans* exposed to bacterial tryptophan metabolites, we conducted Gene Ontology (GO) enrichment analysis of 1334 genes being significantly induced at both condition (Trp-K12

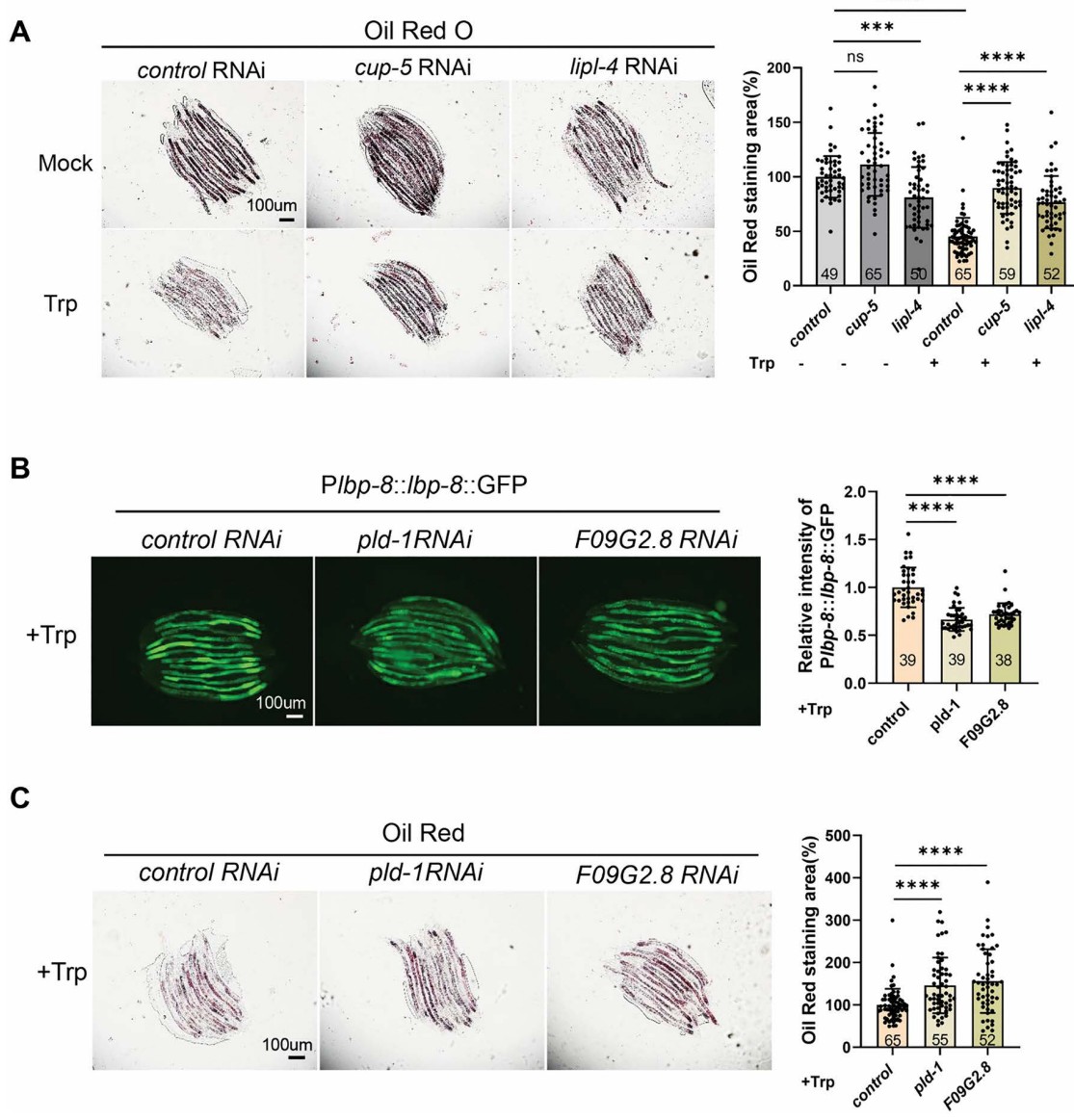

**Fig 4. *E. coli* tryptophan metabolism promotes lysosomal-dependent lipid degradation. (A)** Oil Red O staining and quantification of total lipid levels in wild-type L4 worms treated with control, *cup-5*, or *lipl-4* RNAi, grown on NGM plates with or without 10 mM tryptophan. Scale bar, 100 μm. **(B)** Representative fluorescence images and quantification of the P*lbp-8*::GFP reporter in wild-type L4 worms with control, *pld-1*, or *F09G2.8* RNAi, cultured on NGM + 10 mM tryptophan. Scale bar, 100 μm. **(C)** Oil Red O staining and quantification of lipid content in wild-type L4 worms with control, *pld-1*, or *F09G2.8* RNAi, cultured on NGM + 10 mM tryptophan. Scale bar, 100 μm. Data represent mean ± SD. The number of animals analyzed is indicated. ****$p < 0.0001$,***$p < 0.001$; ns, no significance (ANOVA with multiple-comparisons correction). All experiments were performed independently at least three times. The data underlying this Figure can be found in S1 Data.

versus Mock-K12, Trp-K12 versus Trp-tnaA) (S4B Fig). GO enrichment showed significant enrichment in categories such as "monooxygenase activity", and "oxidoreductase activity" which involved in mitochondria metabolsim (S7A Fig).

During lipid catabolism, lysosomal lipases hydrolyze stored TGs into free fatty acids (FFAs), which are subsequently processed via mitochondrial and peroxisomal β-oxidation to generate ATP [31]. Key genes involved in fatty acid

β-oxidation—including acyl-CoA synthetases (ACS) and acyl-CoA dehydrogenases (ACDH)—were upregulated in Trp-K12-fed animals compared to controls (S7B Fig).

To directly assess β-oxidation activation, we generated transgenic worms expressing *acs-2*::GFP under the control of its native promoter (P*acs-2*). We found that expression of P*acs-2:: acs-2*::GFP is increased in Trp-K12-fed animals but not in those fed Trp-tnaA mutants (Fig 5A), indicating bacterial tryptophan metabolism specifically induces *acs-2*.

Enhanced mitochondrial or peroxisomal β-oxidation often elevates reactive oxygen species (ROS), byproducts of oxidative phosphorylation. Using the redox-sensitive dye MitoTracker Red CMXRos—which reflects mitochondrial membrane potential and ROS levels [32,33]—we observed significantly increased ROS in Trp-K12-fed animals (S8A Fig). This effect was absent in tnaA mutant-fed animals (S8A Fig). Consistent with this, oxidative stress-responsive genes, including *sod-3* (superoxide dismutase; Fig 5B) and *gst-4* (glutathione S-transferase; Fig 5C), were upregulated in Trp-K12-fed worms. Crucially, *acs-2* knockout abolished the enhanced lipid degradation in Trp-K12-fed animals (Fig 5D), demonstrating that ACS-2-dependent mitochondrial β-oxidation is required for tryptophan-mediated lipid loss. Notably, RNAi knockdown of lysosomal genes *cup-5* or *lbp-8* under tryptophan supplementation reduced mitochondrial ROS levels, as measured by MitoTracker Red CMXRos (S8C Fig and S8D Fig) and the oxidative stress reporter P*sod-3*::GFP (Fig 5E and 5F).

Previous studies have shown that lysosomal signaling extends life span by modulating mitochondrial activity—particularly through the LIPL-4–LBP-8 pathway, which upregulates mitochondrial β-oxidation to promote lipid metabolism [12] (Fig 5G). Furthermore, lysosomal function is modulated by multiple longevity pathways and is essential for life span extension [11,22]. Our study provides the key discovery that bacterial-derived indole induces the LIPL-4-LBP-8 signaling pathway and activates lysosomal function (Fig 5G). Previous studies have extensively demonstrated that indole, a secondary metabolite of bacterial tryptophan metabolism, significantly impacts *C. elegans* healthspan, stress resistance, and locomotion during aging [34,35]. Consistent with previous reports, we found that exposure to indole (S8E Fig) or culture with *E. coli* K-12 on LB medium plate (S8F Fig) or supplemented with tryptophan (S8F Fig) significantly increases the survival of young adult worms under oxidative stress. Therefore, it is plausible that bacterial tryptophan inputs, specifically indole, positively influence *C. elegans* longevity, oxidative stress responses, and healthspan parameters through the lysosome-regulated LIPL-4-LBP-8 signaling pathway. Investigating this mechanistic connection represents a promising direction for future research in aging study.

### *E. coli* tryptophan metabolites enhance lysosomal function and promote lipid breakdown in hepatocyte cell line

To investigate the conservation of the mechanism by which bacterial tryptophan metabolism activates lysosomal function to promote lipolysis, we first established an experimental model using Huh7 hepatocyte cell line. Cells were treated with either *E. coli* K12 or tnaA mutant culture supernatants grown in LB medium supplemented with tryptophan (Trp-K12-sup, Trp-TnaA-sup), LB medium alone (control) (Fig 6A).

To investigate the acidification of lysosomes, Hub7 cells were stained with LysoSensor Green DND-189 as the dye accumulates in lysosomes resulting in an increased fluorescence intensity upon acidification. The supernatant from Trp-K12-sup (tryptophan-supplemented *E. coli*) cultures significantly increased lysosomal acidity compared to Trp-TnaA-sup (supernatant from *E. coli* tnaA mutant) (Fig 6B). This effect was abolished by Bafilomycin A1 (BafA1), an inhibitor of lysosomal V-ATPase (Fig 6B). This indicates that bacterial tryptophan metabolism increased lysosomal acidity in Huh7 hepatocyte cell line.

Next, we evaluated lysosomal degradation activity using MagicRed staining—a substrate that emits red fluorescence upon cleavage by Cathepsin B. The supernatant from Trp-K12-sup (tryptophan-supplemented *E. coli*) cultures significantly increased red fluorescence compared to Trp-TnaA-sup (supernatant from *E. coli* tnaA mutant) (Fig 6C). This effect was abolished by BafA1, an inhibitor of lysosomal V-ATPase (Fig 6C). This suggests that bacterial tryptophan metabolism increased lysosomal degradation activity. Cysteine cathepsins are responsible for driving proteolytic degradation within the lysosome [36,37]. Western blot analysis of endogenous cathepsin D (CTSD) processing revealed elevated levels of

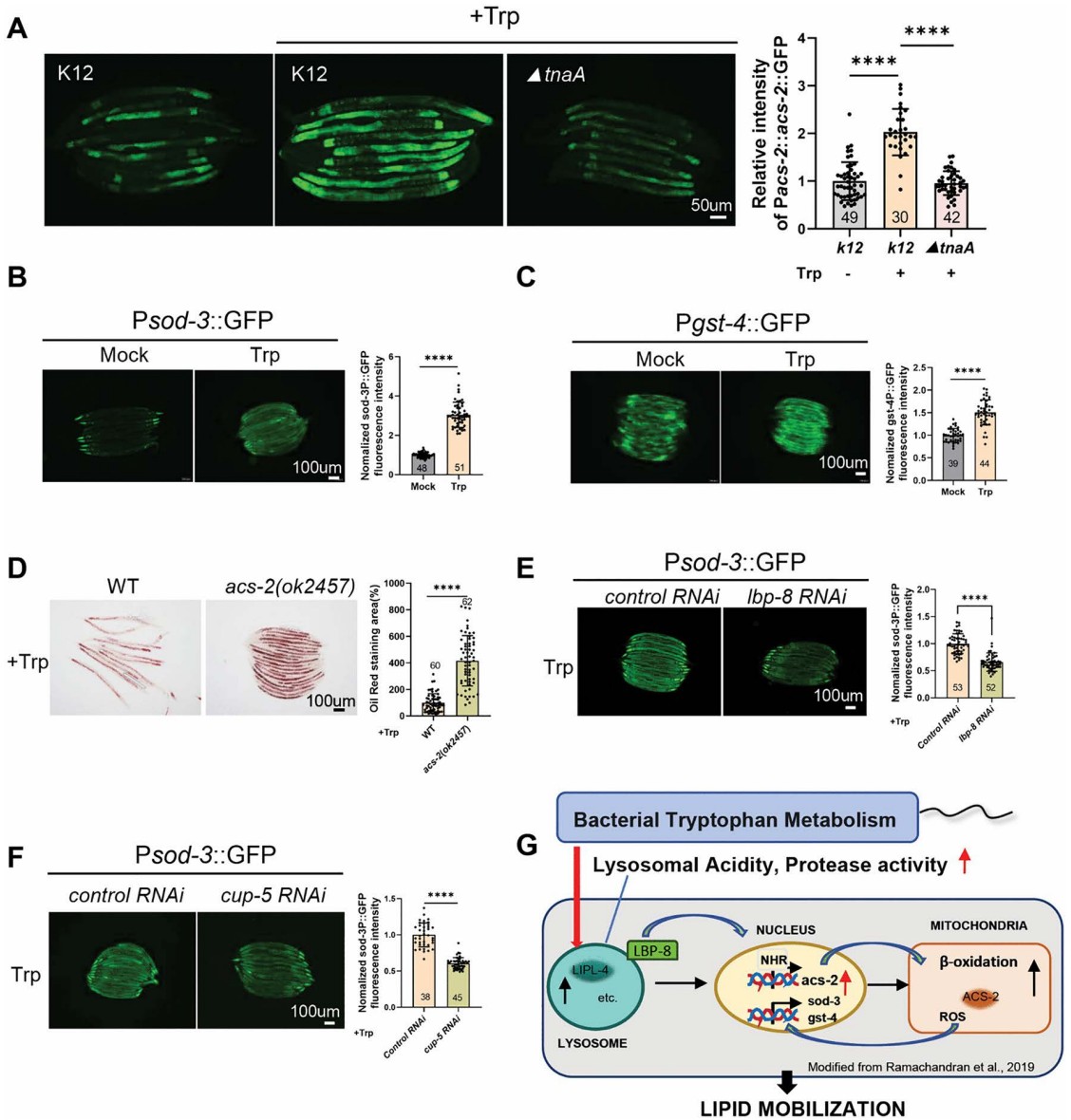

**Fig 5. _E. coli_ tryptophan metabolism enhances mitochondrial β-oxidation via lysosomal activation. (A)** Representative fluorescence images and quantification of the P*acs-2::acs-2::*GFP reporter in L4 worms fed on NGM plates seeded with wild-type _E. coli_-K-12 (control) or with _E. coli_-K-12/ΔtnaA plus 10 mM tryptophan for 60 h. The number of animals analyzed is indicated. Scale bar, 50 μm. **(B)** Representative images and quantification of P*sod-3::*GFP in L4 worms grown on NGM with or without 10 mM tryptophan, seeded with _E. coli_-K-12. The number of animals analyzed is indicated. Scale bar, 100 μm. **(C)** Representative images and quantification of P*gst-4::*GFP in L4 worms grown on NGM with or without 10 mM tryptophan, seeded with _E. coli_-K-12. The number of animals analyzed is indicated. Scale bar, 100 μm. **(D)** Oil Red O staining and quantification of lipid content in wild-type and _acs-2(ok2457)_ mutant L4 worms fed NGM + 10 mM tryptophan, seeded with _E. coli_-K-12. The number of animals analyzed is indicated. Scale bar, 100 μm. **(E)** Representative images and quantification of P*sod-3::*GFP in worms subjected to control or _lbp-8_ RNAi on NGM + 10 mM tryptophan. Scale bar, 100 μm. **(F)** Representative images and quantification of P*sod-3::*GFP in worms subjected to control or _cup-5_ RNAi on NGM + 10 mM tryptophan. The number of animals analyzed is indicated. Scale bar, 100 μm. **(G)** Model of the lysosome–LBP-8–mitochondrial signaling axis by which bacterial tryptophan metabolism stimulates lipid β-oxidation through lysosomal activation. Previous studies have shown that lysosomal signaling extends life span by modulating mitochondrial activity—particularly through the LIPL-4–LBP-8 pathway, which upregulates mitochondrial β-oxidation to promote lipid metabolism. Data represent mean ± SD. All statistical analyses were performed using unpaired two-tailed Student _t_ test. ****$p < 0.0001$. All experiments were performed independently at least three times. The data underlying this Figure can be found in S1 Data.

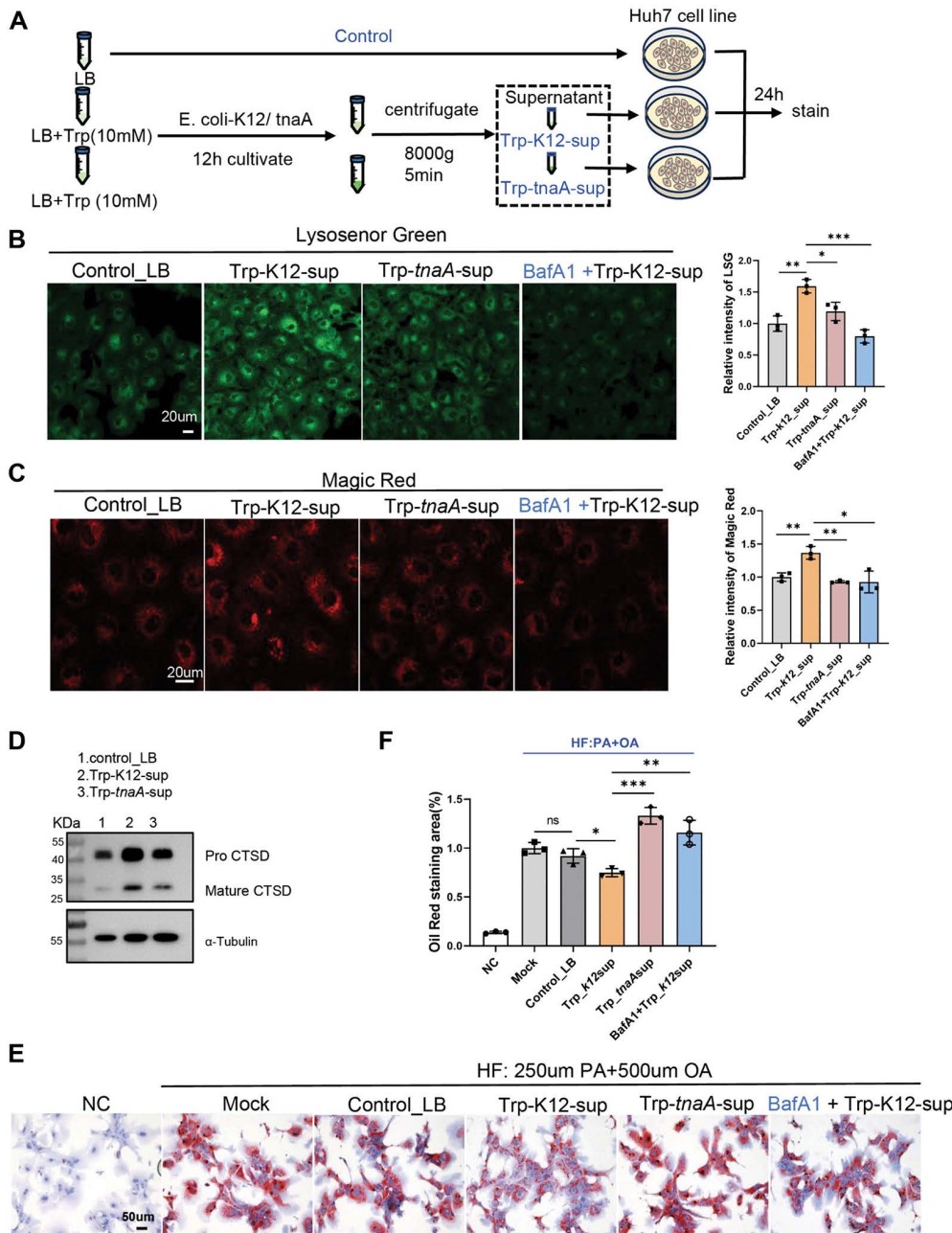

**Fig 6. Bacterial tryptophan metabolites enhance lysosomal function and promote lipid breakdown in hepatocyte cell line. (A)** Schematic of in vitro treatments: Huh7 cells were incubated with *E. coli* K12 or tnaA mutant culture supernatants prepared with tryptophan supplementation (Trp-K12-sup, or Trp-tnaA-sup), or LB medium alone (LB). **(B)** Confocal micrographs and quantification of LysoSensor Green fluorescence in Huh7 cells treated as in **(A)**. Cells were first pre-incubated with 100 nM Bafilomycin A1 (BafA1) for 12 hours to acutely inhibit lysosomal acidification. The statistical analyses used independent experiments as biological replicates (n = 3). For each treatment within an experiment, >45 cells were quantified.Scale bar, 20 μm. **(C)** Confocal micrographs and quantification of Magic Red cathepsin activity in Huh7 cells treated as in **(A)**. Cells were first pre-incubated with 100 nM Bafilomycin A1 (BafA1) for 12 hours to acutely inhibit lysosomal acidification. The statistical analyses used independent experiments as biological replicates (*n* = 3). For each treatment within an experiment, >45 cells were quantified. Scale bar, 10 μm. **(D)** western blot analysis of endogenous cathepsin D (CTSD) in Huh7 cells treated with tryptophan-supplemented *E. coli* K-12 supernatant (Trp-k12 sup), tryptophan-supplemented *E. coli* Δ*tnaA* supernatant (Trp-tnaA sup), or LB control. Both precursor and mature CTSD bands are shown. **(E, F)** Representative Oil Red O staining (E) and quantification of lipid accumulation (F) in primary B6J hepatocytes co-treated for 24 h with high fatty acids [palmitic acid (PA) and oleic acid (OA) (250uM PA, 500uM OA)] and supernatants from Trp-supplemented K-12, *or* LB medium alone (LB). The statistical analyses (F) used independent experiments as biological

replicates ($n = 3$). For each treatment within an experiment, ≥35 cells were quantified. Scale bar, 50 µm. Data represent mean ± SD. All statistical analyses were performed using unpaired two-tailed Student $t$ test. ***$p < 0.001$, **$p < 0.01$, *$p < 0.05$, ns, no significance. All experiments were performed independently at least three times. The data underlying this Figure can be found in S1 Data.

both precursor and mature forms in cells treated with tryptophan-containing supernatants from *E. coli*-K12 (Trp-K12-sup), but not from *E. coli*-tnaA mutant (Trp-tnaA-sup) (Fig 6D). This data further indicated that bacterial-tnaA mediated tryptophan metabolism increased lysosomal degradation activity in Huh7 hepatocyte cell.

To evaluate whether bacterial tryptophan metabolism promotes lipid breakdown, primary mouse hepatocytes were co-treated with palmitic acid/oleic acid (250 µM PA, 500 µM OA) and supernatants from *E. coli* K12 (Trp-K12-sup) or tnaA mutant (Trp-tnaA-sup) cultures. ORO staining revealed significantly reduced lipid accumulation in cells exposed to K12 supernatants, whereas tnaA mutant supernatants showed no difference from controls (Fig 6E and 6F), indicating the role of bacterial tryptophan metabolism in enhancing lipid metabolism in liver cells. Crucially, this lipid-lowering effect was abolished by BafA1 treatment (Fig 6E and 6F), demonstrating that bacterial tryptophan metabolism-mediated lipid reduction requires functional lysosomes. Collectively, our data demonstrate that bacterial tryptophan metabolites, particularly indole, enhances lysosomal activity, promoting lipid degradation in hepatocyte cell line.

### Indole-mediated lipid reduction requires lysosomal function in hepatocytes

To determine whether the bacterial tryptophan metabolite indole promotes lysosomal function and lipid reduction, Huh7 hepatocyte cells were treated with indole (0.5 or 1 mM) in the presence or absence of BafA1, an inhibitor of lysosomal V-ATPase (Fig 7A). LysoSensor Green DND-189 staining demonstrated that indole supplementation significantly increased lysosomal acidity compared to control, an effect that was abolished by BafA1 treatment (Fig 7B). Consistently, Magic Red staining revealed that indole supplementation markedly enhanced Cathepsin B activity, which was similarly abolished by BafA1 treatment (Fig 7C). Furthermore, ORO staining of primary mouse hepatocytes co-treated with palmitic acid/oleic acid and indole showed significantly reduced lipid accumulation, and this lipid-lowering effect was completely abolished by BafA1 treatment, demonstrating that indole-mediated lipid reduction requires functional lysosomes (Fig 7D). Collectively, these findings demonstrate that the bacterial tryptophan metabolite indole enhances lysosomal activity, thereby promoting lipid degradation in hepatocytes.

Previous studies have established that bacterial tryptophan metabolites, including indole [38] and indole-3-acetic acid (IAA) [39], alleviate diet-induced hepatic steatosis, underscoring the physiological relevance of the gut bacterial tryptophan metabolites-liver axis in NAFLD and metabolic syndrome [20]. Given these studies have demonstrated the beneficial effects of indole supplementation on hepatic steatosis [38], we chose not to repeat mice experiments demonstrating indole supplementation attenuates HFD-induced liver steatosis. Although these studies have established the physiological relevance of indole in the gut-liver axis in the context of NAFLD and metabolic syndrome, our study provides the first evidence that the bacterial tryptophan metabolite indole activates lysosomal function, which in turn promotes lipid degradation.

To further strengthen this mechanistic link in vivo, we assessed lysosomal function in primary hepatocytes isolated from indole-fed mice. Mice were orally administered indole (50 mg/kg) or vehicle control for 8 weeks, following the dosing regimen established by Ma and colleagues [38] (Fig 7E). Body weight, water intake, and food consumption were recorded weekly. At the end of the treatment period, primary hepatocytes were isolated, and lysosomal acidity and activity were assessed using LysoSensor Green and Magic Red staining, respectively. No significant differences in body weight (weeks 1–8), food intake (weeks 1–6), or water consumption (weeks 1–4 and 6–7) were observed between groups (S9A–S9C Fig). During weeks 7–8, indole treatment slightly reduced food intake and increased water intake at week 8 (S9A–S9C Fig). Importantly, hepatocytes isolated from indole-fed mice exhibited significantly enhanced lysosomal acidity (Fig 7F) and degradation activity (Fig 7G) compared to those from control mice. Furthermore, western blot analysis of endogenous

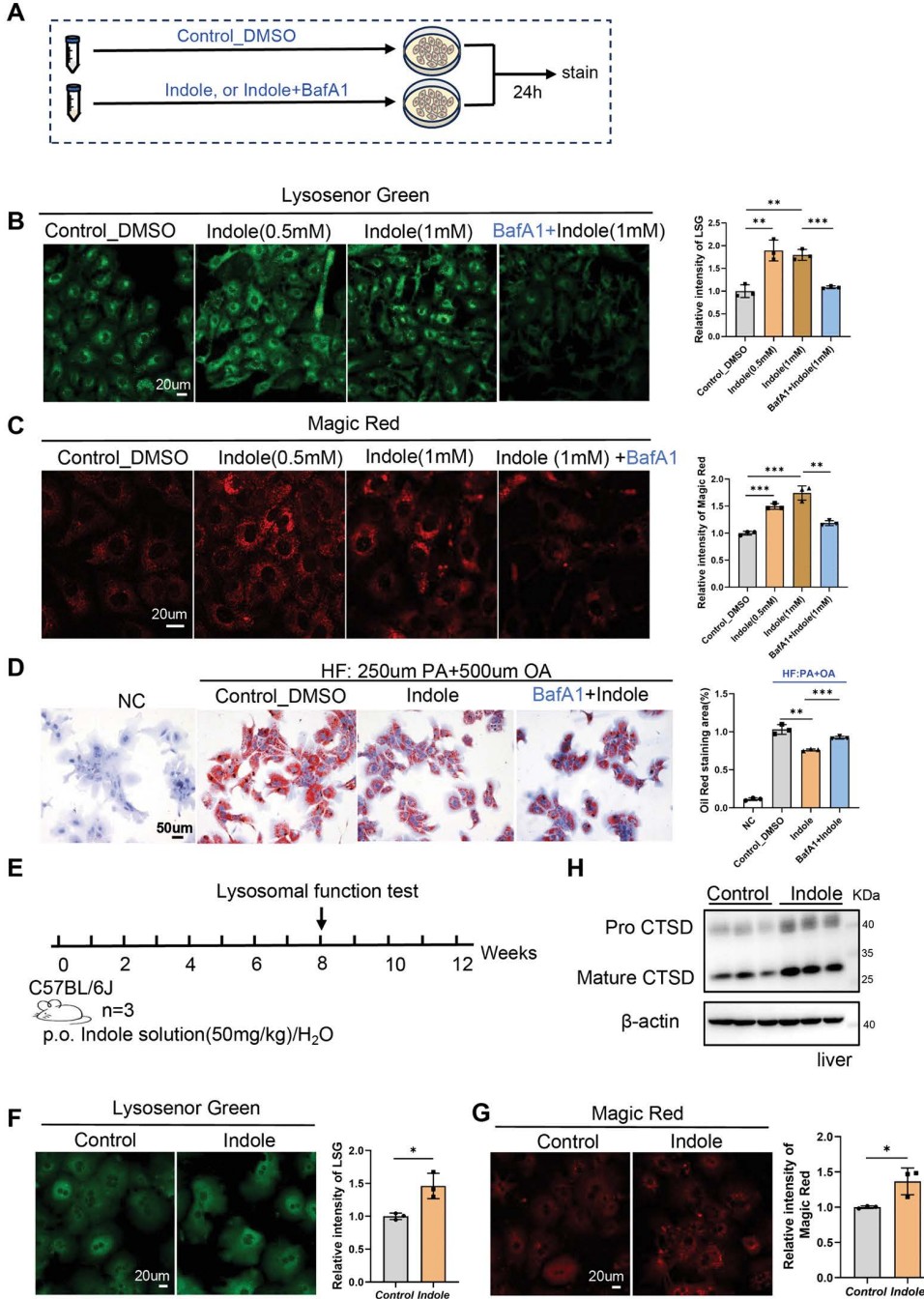

**Fig 7. Bacterial tryptophan metabolite, indole, activates lysosomal function and promotes lipid breakdown. (A)** Schematic of in vitro treatments: Huh7 cells were treated with indole. **(B)** Confocal micrographs and quantification of LysoSensor Green fluorescence in Huh7 cells treated as in **(A)**. The statistical analyses used independent experiments as biological replicates ($n = 3$). For each treatment within an experiment, ≥ 50 cells were quantified. Scale bar, 20 μm. **(C)** Confocal micrographs and quantification of Magic Red cathepsin activity in Huh7 cells treated as in **(A)**. The statistical analyses used independent experiments as biological replicates ($n = 3$). For each treatment within an experiment, ≥40 cells were quantified.Scale bar, 20 μm. **(D)** Representative Oil Red O staining and quantification of lipid accumulation in primary B6J hepatocytes co-treated for 24 h with palmitic acid (PA) and oleic acid (OA) (250uM PA, 500uM OA) with or without indole treatments. The statistical analyses used independent experiments as biological replicates ($n = 3$). For each treatment within an experiment, ≥35 cells were quantified. Scale bar, 50 μm. **(E)** Experimental timeline for the mice study: mice were administered indole (50 mg/kg, orally)or control solution for 8 weeks. Body weight, water intake, and food consumption were recorded weekly ($n = 9$mice/

group). Primary hepatocytes were isolated from 3 mice, and lysosomal acidity and activity were assessed using LysoSensor Green and Magic Red staining. **(F)** Confocal micrographs and quantification of LysoSensor Green fluorescence in primary hepatocytes isolated from 3 mice in **(E)**. Mean fluorescence intensity per mouse was calculated by averaging all cell-level measurements from that animal (>70 cells were quantified per mouse. Statistical comparisons between the control and indole-treated groups were performed using these per-mouse means ($n = 3$ mice per group). Scale bar, 20 μm. **(G)** Confocal micrographs and quantification of Magic Red cathepsin activity in primary hepatocytes isolated from 3 mice in **(E)**. Mean fluorescence intensity per mouse was calculated by averaging all cell-level measurements from that animal (>100 cells were quantified per mouse. Statistical comparisons between the control and indole-treated groups were performed using these per-mouse means ($n = 3$ mice per group). Scale bar, 20 μm. **(H)** western blot analysis of endogenous cathepsin D (CTSD) in the livers of indole-fed mice or controls (3mice/group). Both precursor and mature CTSD bands are shown. Data represent mean ± SD. All statistical analyses were performed using unpaired two-tailed Student $t$ test. \*\*\*$p < 0.001$, \*\*$p < 0.01$, \*$p < 0.05$. All experiments were performed independently at least three times. The data underlying this Figure can be found in S1 Data.

CTSD processing revealed elevated levels of both precursor and mature forms in the livers of indole-fed mice (Fig 7H). These results demonstrate that oral indole supplementation promotes hepatic lysosomal function in vivo.

In summary, our data demonstrate that bacterial tryptophan metabolite indole (i) enhances lysosomal activity, leading to increased lipid degradation in hepatocytes cell; (ii) enhances lysosomal activity in primary hepatocytes isolated from indole-fed mice.

## Discussion

Our study reveals that bacterial tryptophan metabolite indole activates lysosomal function and promotes lipid catabolism in *C. elegans*, and that this effect is recapitulated by indole treatment in mammalian hepatocytes (Fig 8). In *C. elegans*, we show that indole generated via *E. coli* tryptophanase TnaA enhances lysosomal acidification, proteolytic activity, which in turn induces expression of the lipid chaperone LBP-8, stimulates mitochondrial β-oxidation, and depletes lipid stores. This regulatory mechanism is evolutionarily conserved: in mammalian hepatocytes, indole treatment similarly enhances lysosomal function and attenuates lipid accumulation. These findings align with previous reports demonstrating that gut microbiota-derived tryptophan metabolites, such as indole and IAA [20,38,39], protect against diet-induced hepatic steatosis, reinforcing the functional significance of microbial tryptophan metabolism in NAFLD and metabolic syndrome. Collectively, our work establishes lysosomal activation as a previously unrecognized mechanism through which indole, a major product of bacterial tryptophan metabolism, regulates lipid homeostasis in both *C. elegans* and mammalian hepatocytes

Indole production is widespread among bacteria, with over 85 species capable of producing this metabolite, including many Enterobacteriaceae, pathogenic *E. coli* strains, *Shigella* species, *Enterococcus faecalis*, and *Vibrio cholerae* [40]. Notably, studies characterizing the native microbiome of *C. elegans* from natural habitats have consistently identified Enterobacteriaceae as the predominant bacterial group colonizing the nematode gut, alongside *Pseudomonas*,

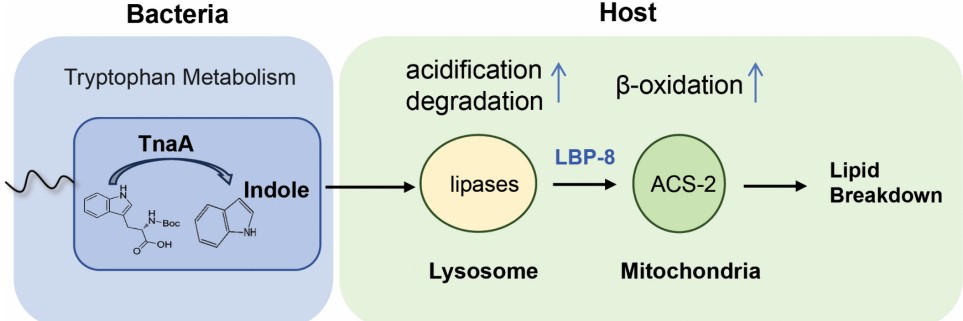

**Fig 8. Proposed model.** In *C. elegans*, *E. coli* TnaA–mediated tryptophan metabolism generates indole, which enhances lysosomal function, induces LBP-8 expression, and activates β-oxidation to promote lipid degradation. In mammalian hepatocytes, indole similarly enhances lysosomal function and reduces lipid accumulation, suggesting a conserved downstream mechanism of indole action.

*Stenotrophomonas*, *Ochrobactrum*, and *Sphingomonas* [41,42]. Given that Enterobacteriaceae include many indole-producing species, *C. elegans* is naturally and frequently exposed to bacterially-derived indole in its environment. The natural association between indole-producing Enterobacteriaceae and *C. elegans* suggests that bacterially-derived indole may serve as a physiologically relevant signal modulating host lipid metabolism through lysosomal activation in this nematode.

The interaction between gut microbiota and lysosomes differs from the evasion strategies employed by intracellular pathogens. Intracellular pathogens such as *Mycobacterium tuberculosis* and *Salmonella* subvert host lysosomal pathways to evade immune defenses by impairing acidification and trafficking [43–45]. In contrast, the ability of commensal gut bacteria to modulate lysosomal function has remained unexplored. Our evidence that commensal bacteria activate lysosomes via tryptophan metabolism (through indole) is based primarily on studies in *C. elegans* with live bacteria. In mammals, findings are limited to experiments supplementing with the bacterial tryptophan metabolite, indole. Whether commensal microbes actively modulate host lysosomal function through this metabolic pathway in the mammalian gut remains to be determined.

Clinical and preclinical studies reveal that metabolic syndrome is associated with diminished microbial production of AhR agonists (including IAA, indole), leading to reduced activation of AhR-dependent pathways that are essential for metabolic health [20]. This deficiency in AhR ligand production directly correlates with disease severity, highlighting a causal link between disrupted bacterial tryptophan metabolism and metabolic dysfunction. Despite these advances, how microbial tryptophan metabolites interface with lysosomal pathways to regulate host metabolism (lipid homeostasis) remained unexplored. Our discovery that bacterial tryptophan metabolism activates lysosomal function, a key regulator of lipid catabolism, and that this activation promotes lipid breakdown in both cellular and animal models provide a mechanistic link between microbial tryptophan metabolism and host metabolic health. This suggests that restoring or enhancing bacterial tryptophan metabolism could be a promising therapeutic strategy to reactivate lysosomal lipolysis and ameliorate lipid-related pathologies.

Beyond lipid metabolism, lysosomes are increasingly recognized for their diverse roles in cellular processes, including nutrient sensing, autophagy, aging, and tissue regeneration [3,11,21]. Given the broad involvement of lysosomes in maintaining cellular homeostasis, the finding that bacterial tryptophan metabolite, indole, can activate lysosomes suggest potential implications beyond lipid metabolism. Future studies could explore whether this pathway plays a role in other physiological contexts regulated by lysosomal function, such as longevity, inflammation, and neurodegenerative diseases.

## Limitations of the study

While our study establishes that bacterial tryptophan metabolite, indole, activates lysosomal function to promote lipid breakdown, several limitations merit attention. First, the molecular mechanisms by which host organisms' sense bacterial tryptophan metabolite to activate lysosomal pathways remain unresolved. We propose future genome-wide genetic screens using the P*lbp-8*::GFP reporter to identify host factors mediating this recognition. Second, while lysosomal activation was inferred indirectly in primary hepatocytes isolated from indole-fed mice, technical limitations precluded direct measurement of lysosomal acidification in hepatic tissues. Development of pH reporters (e.g., NUC-1::pHTomato) for in vivo lysosomal tracking in murine models will bridge this gap. Third, although the *C. elegans* data show that bacterial tryptophan metabolism via TnaA promotes lysosomal function and lipid degradation, the mammalian findings are based on purified indole or bacterial supernatants. Whether bacterial tryptophan metabolism in the mammalian gut generates sufficient indole to produce similar effects in vivo remains unresolved and will require testing in gnotobiotic mice colonized with wild-type or *tnaA*-deficient bacteria. Addressing these gaps will elucidate the molecular mechanisms through which host organisms detect bacterial tryptophan metabolites and orchestrate lysosomal functional activation, thereby advancing our understanding of microbiota-regulated metabolic homeostasis.

## Materials and methods

### Ethics statement

All mice were housed under standard conditions in the animal core facility of Yunnan University. Animal protocols were approved by the Institutional Animal Care and Use Committee (IACUC) of Yunnan University (No. YNU20220262).

### Animals

Male C57BL/6J mice (6–8 weeks old, 18–20 g) were obtained from the Yunnan University Animal Center. Animals were group-housed in a pathogen-free facility under a 12 h light/12 h dark cycle with free access to water and food.

### *C. elegans* strains and maintenance

Unless otherwise noted, nematodes were maintained at 20 °C on standard nematode growth medium (NGM) agar plates seeded with *E. coli* OP50 or *E. coli*-K12.

1. The following strain/alleles was obtained from *Caenorhabditis Genetics Center* (CGC):

   a. N2;

   b. VC4077: *lbp-8(gk5151);*

   c. RB1899: *acs-2(ok2457);*

   d. CF1553: *sod-3p::GFP(muIs84);*

   e. CL2166: *gst-4p::GFP::NLS(dvIs19);*

   f. LIU1: *dhs-3p::dhs-3::GFP(ldrIs1);*

2. The following strain/alleles was obtained from Dr. Xiaochen Wang lab and Dr. Chonglin Yang lab:

   a. XW5399: $P_{ced-1}NUC-1::CHERRY(qxIs257)$;

   b. XW19180: $P_{hs}NUC-1::pHTomato(qxIs750)$;

3. The following strains were generated in our lab:

   a. YNU623: *Plbp-8::GFP;Podr-1::RFP(ylfIs46)*;

   b. YNU624: *Plbp-8::lbp-8::GFP;Podr-1::RFP(ylfIs47)*;

   c. YNU606: *Ex[acs-2P::acs-2::GFP;rol-6](ylfEx325)*;

### Bacterial strains

Bacterial strains (*E. coli* OP50, *E. coli* K-12 BW25113, Keio collection mutants, and *E. coli* HT115) were grown overnight in LB medium at 37 °C. Cultures were then spread onto standard NGM plates or the specialized agar plates described in this study.

### Cell culture

HUH7 cells were maintained at 37 °C in DMEM (VivaCell C3113-0500) supplemented with 10% fetal bovine serum (FBS; VivaCell C04001-500) and penicillin–streptomycin (VivaCell C3421-0100). Primary hepatocytes, isolated from C57BL/6

mouse livers, were cultured at 37 °C in Williams' E medium (Thermo Fisher Scientific 12551032) containing 10% FBS and penicillin–streptomycin. Cells were allowed to adhere fully before any experimental treatments.

## Generation of transgenic strains

P*lbp-8*::*lbp-8*::GFP reporter: The *lbp-8* promoterand genomic *lbp-8* coding sequence (933 bp) were cloned into pPD49.26-GFP. The injection mix contained 10 ng/µl P*lbp-8*::*lbp-8*::GFP and 50 ng/µl *odr-1*p::RFP as a co-injection marker.

P*lbp-8*:: GFP reporter: The *lbp-8* promoter (490 bp) was inserted into pPD49.26-GFP. The injection mix comprised 10 ng/µl P*lbp-8*:: GFP and 50 ng/µl *odr-1*p::RFP.

P*acs-2*::*acs-2*::GFP reporter: The *acs-2* promoter (3,972 bp) and genomic *acs-2* sequence were cloned into pPD49.26-GFP. The injection mix contained 10 ng/µl P*acs-2*::*acs-2*::GFP and 50 ng/µl rol-6(su1006) as a transformation marker.

## Preparation of worm food with various treatments

Preparation of *E. coli* on NGM plate: standard overnight culture of *E. coli* (wild-type: OP50, K12, or tnaA mutant) grown LB broth was spread onto each NGM plate.

Preparation of *E. coli* on LB plate: standard overnight culture of *E. coli* (wild-type: OP50, K12, or tnaA mutant) grown LB broth was spread onto each LB plate.

Preparation of tryptophan-supplemented NGM plates: Dissolve tryptophan (BBI A601911-0050) into standard nematode growth medium (NGM) to a final concentration of 10 mM, then autoclave. Cool the medium to approximately 55 °C, pour into Petri dishes, and allow the agar to solidify. Once set, seed each plate with an overnight culture of *E. coli* (wild-type: OP50, K12, or tnaA mutant). Leave the plates at room temperature for at least 12 hours to allow the bacterial lawn to establish and dry before use in nematode assays.

Preparation of Heat-killed *E. coli* on NGM plates: Following an established protocol to prepare heat-killed *E. coli* [46]. A standard overnight culture of *E. coli* OP50, *E. coli* K12, grown in LB broth was concentrated to 1/10 vol and was then heat-killed at 80 °C for 120 min. About 150 µl of the heat-killed bacteria was spread onto each 3.5 cm NGM plate with or without tryptophan supplementation.

Preparation of indole-supplemented NGM plates: Indole (Sigma, I3408) was initially dissolved in dimethyl sulfoxide (DMSO, Sangon Biotech, A100231) to generate a 0.5M or 1M stock solution, which was subsequently filter-sterilized through 0.22-µm filter membranes. Aliquots of the sterile stock were then incorporated into autoclaved nematode growth medium (NGM) at a 1:1000 (v/v) ratio to a final concentration of 0.5 mM or 1 mM before pouring onto NGM agar plates.

## Fluorescence intensity measurement

Synchronized L1-stage reporter worms (P*lbp-8*::GFP, P*lbp-8::lbp-8::*GFP, P*sod-3*::GFP, or P*acs-2*::*acs-2*::GFP) were cultured in the indicated media at 20 °C until the L4 stage. For imaging, worms were immobilized in 10 mM levamisole and mounted on agarose pads. Fluorescence was captured under excitation on an Olympus BX53 microscope fitted with a DP80 camera. Whole worm fluorescence intensity was quantified in ImageJ, with at least 30 worms analyzed per reporter strain.

## Quantification of the nucleation rate of LBP-8 protein

Synchronized L1-stage P*lbp-8::lbp-8::*GFP reporter worms were cultured in the specified media at 20 °C until the L4 stage. Individual animals were imaged on a Zeiss LSM 900 laser-scanning confocal microscope. Nuclear and cytoplasmic GFP intensities were quantified using Zeiss ZEN Blue software to calculate the nucleus-to-cytoplasm fluorescence ratio. At least 20 worms were analyzed per condition.

## *E. coli* Keio collection screen

The *E. coli* Keio collection, a library of single-gene knockout mutants [47], was used to screen for genes affecting the host's *lbp-8* expression. For the screen, individual Keio strains were grown overnight at 37 °C in LB medium

supplemented with 50 μg/mL kanamycin. A 150 μL aliquot of each culture was spread onto 35 mm LB agar plates. Synchronized L1-stage *C. elegans* carrying the P*lbp-8*::GFP reporter were then transferred onto these plates and incubated at 20 °C for 60 h. Mutants that consistently reduced the P*lbp-8*::GFP expression were selected for follow-up validation assays.

**RNAi treatment in *C. elegans***

All RNAi-by-feeding experiments used *E. coli* HT115 clones from the MRC RNAi Library [48] or the ORF-RNAi Library [49]. RNAi bacterial cultures were grown overnight at 37 °C in LB medium supplemented with 50 μg/mL carbenicillin, then seeded onto NGM agar plates containing 1 mM IPTG and 50 μg/mL carbenicillin. Synchronized L1-stage worms were transferred onto these RNAi plates and maintained at 20 °C. Worms were allowed to develop either to the L4 stage or the next generation, depending on the specific experimental requirements.

**RNA-seq preparation and analysis**

1. Preparation of *C. elegans* samples for RNA-seq

RNA sequencing was conducted with triplicate biological replicates, each representing independent experimental batches.
Group 1 (NGM-based cultivation): Synchronized L1 wild-type animals grown on NGM plate (seeded with *E. coli* K12 or tnaA mutant) supplemented with or without tryptophan.
Group 2 (LB-based cultivation): Synchronized L1 wild-type animals grown on LB plate (seeded with *E. coli* K12 or tnaA mutant) to L4 stage.
All cultures were maintained at 20 °C until L4 developmental stage (48–60 h post-synchronization), as determined by vulval morphology. Animals were then flash-frozen in TRIzol (Invitrogen) for RNA extraction.

2. RNA sequencing and data processing

All sequencing was performed at Biomarker Technologies (BMKGENE). For the RNA sequencing assay, the reference genome index was built using HISAT2 v2.0.5, and paired-end clean reads were aligned to the reference genome with HISAT2 v2.0.5. Differential gene expression analysis was conducted using DESeq2, with p-values adjusted using the Benjamini-Hochberg method [50] to control the false discovery rate. Genes with an adjusted *p*-value ≤0.05 were considered differentially expressed. GO enrichment analysis and KEGG pathway enrichment were performed using the clusterProfiler package, with GO terms and KEGG pathways considered significantly enriched if *p* < 0.05.

**Oil Red O (ORO) staining**

***C. elegans*:** Young adult worms were collected, washed twice with M9 buffer, and fixed in 0.5% paraformaldehyde (PFA) (BBI, A500684-0500) for 30 min following three freeze-thaw cycles in liquid nitrogen. Worms were then washed with M9 to remove residual PFA and incubated with 60% ORO working solution at room temperature in the dark for 30 min. The 60% ORO working solution was freshly prepared by diluting ORO stock solution (Sigma, #O1391) with water, followed by rocking and filtration through a 0.22-μm filter. After incubation, worms were washed three times with M9 and mounted onto 2% agarose pads for imaging.
**Mammalian cells:** Primary hepatocytes, isolated from C57BL/6 mouse livers, were cultured at 37 °C in Williams' E medium (Thermo Fisher Scientific 12551032) containing 10% FBS and penicillin–streptomycin. Cells were allowed to adhere fully before treatments. For Fig 6E, the primary mouse hepatocytes were co-treated with palmitic acid (PA) and oleic acid (OA) (250uM PA, 500uM OA) and supernatants from *E. coli* (K12, or tnaA) LB cultures+tryptophan (vol of bacterial supernatants: vol of cell cultures = 1:10) or only LB medium for 24 hours. For Fig 7D, the primary mouse hepatocytes

 

were co-treated for 24 h with PA and OA (250uM PA, 500uM OA) with or without indole treatments (indole concentration: 0.5mM)

Then, cells were washed three times with PBS after removing the culture medium, then fixed with 4% PFA for 1 hour. Following fixation, cells were washed three times with PBS for 5 min each and soaked in 60% isopropanol for 2 min before incubation with 60% ORO working solution at 37 °C for 15 min. After staining, cells were washed three times with PBS, and nuclei were stained with hematoxylin (Servicebio, #G1004). Coverslips were mounted using 80% glycerin before imaging.

Prepared samples were examined using an Olympus BX53 microscope equipped with a DP80 camera. To quantify ORO staining intensity, mean intensity was measured using ImageJ software.

### Detection of oxidative stress in *C. elegans*

MitoTracker Red CMXRos is used as an indicator of mitochondrial membrane potential. Under conditions of excessive mitochondrial ROS production, oxidative damage leads to a loss of membrane potential, resulting in reduced dye retention and diminished fluorescence intensity [51].

MitoTracker Red CMXRos staining was performed according to a previously published method with some modifications [33]. A 1 mM stock solution of MitoTracker Red CMXRos was prepared in DMSO. To prepare the staining solution, 0.4 μL of the stock solution was added to 50 μL of M9 buffer and mixed thoroughly. Approximately 30–40 worms were then transferred into the solution and incubated in the dark for 5–10 min. After staining, the worms were placed on agar plates at 20 °C for a 30-min recovery period. Fluorescence imaging was performed using an Olympus BX53 microscope equipped with a DP80 camera, capturing signals in the RFP fluorescence channel.

### Quantification of lysosomal morphology

NUC-1::mCherry reporter worms at different ages (days 1, 3, and 6) were imaged using laser scanning confocal microscopy (LSM 900, Carl Zeiss). Serial optical sections were analyzed, and the relative area of NUC-1::mCherry-positive vesicular lysosomes and the length of per lysosomal tubules were quantified per unit area 31x43 $um^2$ using Zeiss ZEN Blue software. (The relative rare of vesicular lysosome = Sum of vesicular lysosome areas in unit area/unit area $31 \times 43$ $μm^2$). The length of tubular lysosomes were randomly quantified at least 10 in each unit area. At least 10 worms were scored in each group at each age.

### Quantification of NUC-1::pHTomato intensity

Day 2 adult *C. elegans* expressing P*hs*NUC-1::pHTomato were incubated at 33 °C for 30 min, followed by a 24-hour recovery period at 20 °C before analysis. Worms were imaged using laser scanning confocal microscopy (LSM 900, Carl Zeiss). The average pHTomato fluorescence intensity per lysosome was quantified using Zeiss ZEN Blue software. At least 20 worms were analyzed for each condition.

### *E. coli* tryptophan metabolite treatment in cell lines

*E. coli*-K12 or *E. coli*-tnaA strains were cultured overnight in LB medium supplemented with or without 50 mM tryptophan. LB medium alone served as the vehicle control. Following overnight incubation, bacterial cultures were centrifuged at 4,000*g* for 15 min to collect supernatants, which were then filter-sterilized through 0.22-μm filter membranes.

For cell treatments, cells at approximately 50%–60% confluence were exposed to the following group for a duration of 24 hours.

*E. coli*-K12 culture supernatants derived from LB medium supplemented with tryptophan (Trp-K12-sup); vol of supernatants: vol of cell cultures = 1:10.

E. coli-tnaA culture supernatants derived from LB medium supplemented with tryptophan (Trp-tnaA-sup); vol of supernatants: vol of cell cultures = 1:10.

E. coli-K12 culture supernatants from LB medium lacking tryptophan (K12-sup); vol of supernatants: vol of cell cultures = 1:10.

Sterile LB medium alone (control); vol of LB: vol of cell cultures = 1:10.

### Indole treatment in cell lines

Indole (Sigma, I3408) was initially dissolved in dimethyl sulfoxide (DMSO, Sangon Biotech, A100231) to generate a 0.5M or 1M stock solution, which was subsequently filter-sterilized through 0.22-µm filter membranes. Aliquots of the sterile stock were then incorporated into the cell culture medium at a 1:1000 (v/v) ratio to a final concentration of 0.5 mM or 1 mM. Adding DMSO solution with the same volume as control into cell culture medium. Cells at approximately 50%–60% confluence were subjected to treatment for a duration of 24 hours.

### Survival analysis of animals under tert-butyl hydroperoxide (t-BOOH)

For t-BooH (Aladdin, B106035), the compound was added to molten agar immediately before pouring onto 60 mm NGM agar plates(to a final concentration of 10 mM). Plates were dried and seeded with 200 µl E. coli OP50. Nematodes were cultured in various conditioned plates at 20°C until day-1 stage, and then transferred to NGM with 10 mM tBuOOH added. A typical experiment consisted of three assay plates of each condition, each assay plate containing a maximum of 60 nematodes.All experiments were repeated at least twice, yielding the same results. Animals were scored for survival per 12 h.

### LysoSensor Green staining in cell lines

The fluorescent probe LysoSensor Green DND-189 (Thermo Fisher, L7535) is used to measure lysosomal pH. With a pKa of ~5.2, this dye becomes more fluorescent in acidic environments. To perform staining, the cell culture medium was removed, and cells were washed twice with PBS. The cells were then incubated in 500 µL of DMEM containing 1 µM LysoSensor Green DND-189 at 37 °C for 1–2 min. Fluorescence images were captured using laser scanning confocal microscopy (LSM 900, Carl Zeiss), and fluorescence intensity was quantified using ImageJ software.

### Quantification of NUC-1::CHERRY cleavage

Approximately 200 Day 1 adult worms expressing NUC-1::CHERRY were washed with M9 buffer. Worms were lysed through three cycles of freezing and thawing, followed by boiling with SDS loading buffer. The worm lysate was analyzed by western blot using anti-CHERRY antibodies (Proteintech, 26765-1-AP, 1:3000) and anti-tubulin antibodies (Sigma, T5168,1:5000). The intensities of NUC-1::CHERRY and free CHERRY bands were quantified using ImageJ software. Cleavage efficiency was calculated as the ratio of CHERRY to the total NUC-1::CHERRY and CHERRY signal.

### Magic Red staining

Magic Red stain (Abcam, ab270772) is a cell-permeable dye that enters cells and organelles, where it is cleaved by cathepsin B, generating a red fluorescent signal in functional lysosomes. Magic Red stain was prepared in 250× DMSO stock following the manufacturer's instructions.

For worms staining, dilute the 250× stock solution 1:10 in deionized water (diH$_2$O) to prepare a 25× staining solution. Then, mix 20 µL of the staining solution with 480 µL of M9 buffer and spread it onto a small plate containing 1mL NGM. Allow the plate to dry under a hood. L4-stage C. elegans were transferred onto Magic Red-containing plates and incubated overnight. Fluorescent images were captured using an Olympus BX53 microscope equipped with a DP80 camera, and fluorescence intensity was quantified using ImageJ.

For cell lines staining, the 250× Magic Red stock was diluted 1:10 in diH$_2$O to prepare a 25× staining solution. Then, 20 µL of the staining solution was further diluted in 480 µL of DMEM (20 µL stain+480 µL DMEM = 500 µL). The cell culture medium was removed, and the staining solution was added to the Huh7 cells (1 ml diluted staining solution added into 35 mm confolcal dish), followed by incubation at 37 °C for 45 min. Fluorescence images were captured using laser scanning confocal microscopy (LSM 900, Carl Zeiss), and fluorescence intensity was quantified using ImageJ software.

## Detection of CTSD maturation in mammalian cells

CTSD is a lysosomal aspartic protease belonging to the pepsin superfamily. In human cells, CTSD is initially synthesized as pre-pro-CTSD and undergoes glycosylation in the rough endoplasmic reticulum (ER). The signal peptide is cleaved in the ER, generating pro-CTSD, which is then transported to the Golgi and subsequently to the endosome. Within the lysosome, pro-CTSD is further processed into its mature form (m-CTSD) [52].

Lysates from Huh7 cells were prepared using radioimmunoprecipitation assay (RIPA) buffer. Protein samples (10–30 µg) were separated by 10% SDS-PAGE and transferred onto polyvinylidene fluoride membranes. Membranes were blocked and incubated overnight with primary antibodies against CTSD (Abcam, ab75852, 1:3,000) and β-Actin (Cell Signaling, 4968S,1:5,000). After washing with TBS-Tween and incubation with the appropriate HRP-conjugated secondary antibody anti-Rabbit IgG (ABclonal, AS014, 1:10,000) and goat anti-mouse (Invitrogen, 62-6520, 1:10,000), protein bands were visualized using enhanced chemiluminescence (ECL, Thermo Scientific). The intensities of mature CTSD and pro-CTSD bands were quantified using ImageJ software.

## Animal experiments with indole treatment

Male C57BL/6J mice (6–8 weeks old, 18–20 g) were obtained from the Yunnan University Animal Center. All mice were housed under specific pathogen-free conditions with a 12-h light/dark cycle. Mice were maintained on a standard chow diet throughout the study. Mice were treated with indole (orally, 50 mg/kg, suspension in autoclaved tap water albumin) or autoclaved tap water; Water was provided in standard rodent drinking bottles and changed weekly.

Body weight, food intake, and water consumption were measured weekly at a consistent time point (between 4:00 PM and 6:00 PM) (n = 9 per group).

Mice (n = 3 per group) were euthanized at 8 weeks post-treatment initiation. Primary hepatocytes were then isolated via collagenase perfusion and subsequently cultured in vitro for further analysis. Cultured hepatocytes were stained with LysoSensor Green DND-189 (for assessing lysosomal pH) and Magic Red Cathepsin B substrate (for evaluating protease activity) according to standard protocols.

## Microscopy

The *C. elegans* screening was performed using an Olympus MVX10 dissecting microscope. Fluorescence images were captured with an Olympus BX53 microscope equipped with a DP80 camera. Confocal images were acquired using an inverted Zeiss LSM 880/900 confocal microscope system, fitted with an alpha Plan-Apochromat 63× oil immersion objective lens.

## Statistical analysis

All experiments were performed independently at least three times with similar results. For experiments involving primary hepatocytes or cell lines, the biological replicate (defined as independent mice for primary hepatocyte experiments, or independent experimental runs for cell line experiments) was used as the unit of statistical analysis. Statistical analyses were performed using Student *t* test or one-way analysis of variance (ANOVA). All analyses were conducted using GraphPad Prism software. Staining quantification was carried out using ImageJ and ZEN software. Data are presented as

mean ± SD. *P*-values < 0.05 were considered statistically significant. **p* < 0.05, ***p* < 0.01, ****p* < 0.001, *****p* < 0.0001, and "ns" indicates no significant difference. The Log rank (Mantel-Cox) test was used for statistical analysis of survival assay.

## Supporting information

**S1 Fig. P*lbp-8*::GFP reporter expression on metabolically compromised *E. coli, and* Bacterial growth state on NGM and LB plate. (A)** Representative fluorescence images of the P*lbp-8*::GFP reporter in worms fed with metabolically compromised *E. coli* (treated with ampicillin or UV-killed). "ns" indicates no significant difference (*p* > 0.05, Student *t* test). Data represent mean ± SD. The number of analyzed animals is indicated. Scale bar: 100 μm. **(B)** Bacterial growth state on NGM and LB culture media seed with *E. coli*-K12. **(C)** Fluorescence images and quantification of the P*lbp-8::lbp-8::*GFP reporter in 24h after L1 stage feed on NGM plate or LB plate with wild-type *E. coli* K12 or Heat-Kill *E. coli* K12 (see Methods) or None *E. coli.*Scale bar: 50 μm. Data represent mean ± SD. All statistical analyses were performed using unpaired two-tailed Student *t* test. *****p* < 0.0001, ***p* < 0.01, ns, no significance. All experiments were performed independently at least three times. The data underlying this Figure can be found in S1 Data.
(TIF)

**S2 Fig. Screen of *E. coli* mutants defective in lbp-8 induction on LB plates. (A)** Representative fluorescence images and quantification of the P*lbp-8*::GFP reporter in worms fed with *E. coli* K12 mutants, which significantly suppressed *lbp-8* expression on LB medium. Scale bar: 100 μm. (B) Gene names and funcitonal annotation of 19 *E. coli* mutants. (C) GO Biological Process enrichment for the 19 mutated genes. Data represent mean ± SD. *****p* < 0.0001 (ANOVA with multiple-comparisons correction). All experiments were performed independently at least three times. The data underlying this Figure can be found in S1 Data.
(TIF)

**S3 Fig. Bacterial tryptophan metabolism activates lysosomal function. (A)** *lbp-8* mRNA levels in L4-stage wild-type worms exposed to *E. coli* K12 on NGM (NGM-K12) and *E. coli* K12 or *tnaA* cultured on LB medium (LB-K12/LB-*tnaA*), which was extracted from RNA-seq data. **(B)** Fluorescence images of the P*lbp-8::lbp-8::*GFP reporter in worms exposed to wild-type *E. coli* K12 or *tnaA* mutants on LB medium, along with quantification of nucleus-to-cytoplasm GFP intensity of LBP-8 protein. The number of animals analyzed is indicated. Scale bar: 10 μm. **(C)** *lbp-8* mRNA levels in L4-stage wild-type worms exposed to *E. coli* K12 grown under standard NGM conditions (Mock) or supplemented with 10 mM tryptophan (*E. coli* K12-Trp, *tnaA*-Trp), which was extracted from RNA-seq data. **(D)** Fluorescence images of the P*lbp-8::lbp-8::*GFP reporter in worms exposed to wild-type *E. coli* on NGM medium with or without 10 mM tryptophan, along with quantification of nucleus-to-cytoplasm GFP intensity of LBP-8 protein. The number of animals analyzed is indicated. Scale bar: 10 μm. **(E)** Fluorescence images of the P*lbp-8::lbp-8::*GFP reporter in worms exposed to wild-type *E. coli* on NGM medium with or without 10 mM tryptophan. The number of animals analyzed is indicated. Scale bar: 100 μm. **(F)** Fluorescence images of the P*lbp-8::lbp-8::*GFP reporter in worms exposed to wild-type *E. coli* or heat-killed *E. coli* cultured in LB medium with or without tryptophan. The number of animals analyzed is indicated. Scale bar: 100 μm. **(G)** Fluorescence images and quantification of the P*lbp-8*::*lbp-8*::GFP reporter in L4 animals feed on NGM plate (seeded with *E. coli*- K12) supplemented with1 mM secondary metabolites of bacterial tryptophan (including pyruvate, indole, ammonium chloride and ammonium sulfate). **(H)** High-magnification fluorescence images and quantification of the nuclear-to-cytoplasmic GFP intensity ratio for LBP-8::GFP in L4 larvae on NGM plate (seeded with *E. coli*- K12) with or without indole treatment. Scale bar, 10 μm. **(I)** Fluorescence images and quantification of the DHS-3::GFP reporter in wild-type Adult Day1 fed on NGM plate (seeded with E. *coli*-K12) with or without 0.5mM indole. Number of lipid droplets were quantified per unit area 26 × 26 μm$^2$. The diameter of a minimum of 10 lipid droplets are randomly measured in each worm. At least 16 worms were scored in each group at each age. Scale bar, 5 μm. Data represent mean ± SD. Statistical analyses were performed using unpaired two-tailed Student *t* test (A–F,H–I) or ANOVA with multiple-comparisons correction(G). *****p* < 0.0001, ns,

no significance. All experiments were performed independently at least three times. The data underlying this Figure can be found in S1 Data.
(TIF)

**S4 Fig. Bacterial tryptophan metabolism induce lysosomes-related genes expression.** (A) Schematic diagram illustrating the RNA sequencing sample preparation process, including the three experimental conditions: NGM plates seeded with *E. coli* K12 (Mock-K12), tryptophan-supplemented NGM plates seeded with *E. coli* K12 (Trp-K12), and tryptophan-supplemented NGM plates seeded with *E. coli tnaA* (Trp-*tnaA*). Synchronized L1-stage wild-type animals were grown on these plates until reaching the L4 stage, at which point L4-stage animals were collected for sequencing. (B) Venn diagram showing the overlap between genes differentially expressed in Trp-K12 versus NGM-K12 and in Trp-K12 versus Trp-*tnaA*. (C) KEGG enrichment analysis of differentially expressed genes in Trp-K12 versus NGM-K12. (D) Venn diagram displaying the overlap of lysosomal-related genes, which are induced at both group (Trp-K12 versus Mock-K12, Trp-K12 versus Trp-tnaA). (E) List of overlapping lysosomal-related genes. (F) mRNA levels from RNA-seq of the lysosomal-related genes identified in (D). The data underlying this Figure can be found in S1 Data.
(TIF)

**S5 Fig. Bacterial tryptophan metabolism and indole regulate lysosomal function. (A)** Confocal images of lysosomal morphology in wild-type adults expressing NUC-1::mCherry at days 1, 3, and 6 of adulthood. Animals were maintained on NGM plates seeded with K12 (control) or with K12/Δ*tnaA* plus 10 mM tryptophan. White arrowheads denote vesicular lysosomes; yellow arrowheads indicate distinct lysosomal tubules. The relative area of vesicular lysosomes and the length of per lysosomal tubules were quantified per unit area $31 \times 43$ $\mu m^2$. (the relative rare of vesicular lysosome = Sum of vesicular lysosome areas in unit area/unit area $31 \times 43$ $\mu m^{2)}$). At least 10 worms were scored in each group at each age. Scale bar, 10 μm. **(B)** Representative images and quantitative analysis of lysosomes of wild-type at day-2 adults fed on NGM plate with or without 0.5 mM indole stained by Lysosensor Green. Scale bar, 100 μm. Data represent mean ± SD. Statistical analyses were performed using unpaired two-tailed Student *t* test (B) or ANOVA with multiple-comparisons correction (A). ****$p < 0.0001$, **$p < 0.01$,*$p < 0.05$ ns, no significance. All experiments were performed independently at least three times. The data underlying this Figure can be found in S1 Data.
(TIF)

**S6 Fig. Bacterial tryptophan metabolism promotes lipid degradation in a lysosomal-dependent manner. (A, B)** Representative fluorescence images (A) and quantification (B) of the P*lbp-8*::GFP reporter in wild-type L4 worms treated with control, *cup-5*, *lipl-4*, or *lbp-8* (negative control) RNAi, grown on NGM plates with or without 10 mM tryptophan. The number of animals analyzed is indicated. Scale bar, 100 μm. Data represent mean ± SD.****$p < 0.0001$, "ns" $p > 0.05$ (ANOVA with multiple comparisons correction). Scale bar, 100 μm. The data underlying this Figure can be found in S1 Data. **(C)** Homology alignment of *C. elegans* PLD-1 and F09G2.8 with human PLD-3 and PLD-4 protein sequences. For each row/column, 100% represents the target protein, and percentages indicate sequence similarity relative to the target protein.
(TIF)

**S7 Fig. Gene expression in L4 staged animals grown on NGM plate (seeded with *E. coli* K12 or tnaA mutant) supplemented with or without tryptophan. (A)** Gene ontology (GO) enrichment analysis of 1334 genes being significantly induced at both condition (Trp-K12 vs. Mock-K12, Trp-K12 vs. Trp-tnaA). **(B)** Heatmap showing fold changes in mRNA levels of β-oxidation-related genes in worms fed Trp-supplemented K12 (Trp-K12) versus mock-treated K12 (Mock-K12), and Trp-supplemented Δ*tnaA* (Trp-tnaA) versus Trp-supplemented K-12 (Trp-K12). Fold changes were calculated by dividing each gene's expression level under the indicated conditions. The data underlying this Figure can be found in S1 Data.
(TIF)

**S8 Fig. Bacterial tryptophan metabolism enhances mitochondrial β-oxidation via lysosomal activation to facilitate lipid metabolism. (A)** Representative fluorescence images and quantification of MitoTracker Red CMXRos in L4 worms fed on NGM plates seeded with wild-type K12 (control) or with K12/ΔtnaA plus 10 mM tryptophan. Data represent mean ± SD. The number of animals analyzed is indicated. ***$p < 0.001$;**$p < 0.01$; (Student $t$ test). Scale bar: 100 μm. **(B, C)** Representative microscope images and quantification of MitoTracker Red CMXRos fluorescence in wild-type, *acs-2(ok2457)* (B) or *lbp-8(gk5151)* (C) mutants L4 worms fed NGM + 10 mM tryptophan, seeded with *E. coli*-K12. Data represent mean ± SD. The number of animals analyzed is indicated. ****$p < 0.0001$ (Student $t$ test). Scale bar: 100 μm. **(D)** Represen*t*ative microscope images and quantification of MitoTracker Red CMXRos fluorescence in wild-type worms subjected to control or *cup-5* RNAi on NGM + 10 mM tryptophan. Data represent mean ± SD. The number of animals analyzed is indicated. ****$p < 0.0001$ (Student $t$ test). Scale bar: 100 μm. **(E)** Survival analysis of day-1 adult wild-type animals on NGM plates with 10 mM tert-butyl hydroperoxide (tBuOOH) added. Nematodes were cultured on NGM plates with or without indole seeded with *E. coli* K12 supplementation until day-1 adult and were then transferred to tBuOOH plate for subsequent assays. ****$p < 0.0001$ (Log-rank (Mantel-Cox) test). **(F)** Survival analysis of day-1 adult wild-type animals on NGM plates with 10 mM tert-butyl hydroperoxide (tBuOOH) added. Nematodes were cultured on NGM plates or or LB plates seeded with *E. coli* K12 until day-1 adult and were then transferred to tBuOOH plate for subsequent assays. ****$p < 0.0001$ (Log-rank (Mantel-Cox) test). **(G)** Survival analysis of day-1 adult wild-type animals on NGM plates with 10 mM tert-butyl hydroperoxide(tBuOOH) added. Nematodes were cultured on NGM plates seeded with *E. coli* K12, tryptophan-supplemented NGM plates seed with *E. coli* K12 or *E. coli* tnaA, and all were then transferred to tBuOOH plate for subsequent assays. ****$p < 0.0001$ (Log-rank (Mantel-Cox) test). The data underlying this Figure can be found in S1 Data.
(TIF)

**S9 Fig. The phenotype of mice trated with indole. (A–C)** mice were administered indole (50 mg/kg, orally) or control solution for 8 weeks (*n* = 9/group). Body weight (A), food intake (B), and water intak (C) were recorded weekly. Data represent mean ± SD.**$p < 0.01$; *$p < 0.05$; ns, not significant(ANOVA with multiple comparisons correction). The data underlying this Figure can be found in S1 Data.
(TIF)

**S1 Table. RNA-seq data related to Figs 3 and S4A.**
(XLSX)

**S1 Data. Numerical data for all figures in this study.** This file contains multiple datasheets.
(XLSX)

**S1 Raw Images. Data supplement for Western Blot Full Gel Images.**
(PDF)

## Acknowledgments

We thank the *Caenorhabditis* Genetics Center (CGC) (funded by NIH P40OD010440) for strains; Dr. Xiaochen Wang (SUSTech) and Chonglin Yang (Yunnan University) for sharing strains and suggestions.

## Author contributions

**Conceptualization:** Kenan Zhang, Bin Qi, Zhao Shan.

**Data curation:** Kenan Zhang, Zihan Luo, Bin Qi, Zhao Shan.

**Formal analysis:** Kenan Zhang, Bin Qi, Zhao Shan.

**Funding acquisition:** Bin Qi, Zhao Shan.

**Investigation:** Kenan Zhang, Zihan Luo, Yan Chen, Yan Li, Bin Qi, Zhao Shan.

**Methodology:** Kenan Zhang, Zihan Luo, Yan Chen, Lang Wang, Yanan Liu, Ruizhi Yang, Qian Li, Jiahao Zhao, Bin Qi, Zhao Shan.

**Project administration:** Bin Qi, Zhao Shan.

**Resources:** Kenan Zhang, Bin Qi.

**Supervision:** Bin Qi, Zhao Shan.

**Validation:** Kenan Zhang, Zihan Luo, Bin Qi, Zhao Shan.

**Visualization:** Kenan Zhang, Bin Qi, Zhao Shan.

**Writing – original draft:** Kenan Zhang, Bin Qi, Zhao Shan.

**Writing – review & editing:** Kenan Zhang, Bin Qi, Zhao Shan.

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
