## [Editor Report · Decision Letter 0]

7 Aug 2025

Dear Bin,

I hope all is well. Thank you for submitting your manuscript entitled "Microbial Tryptophan Metabolism Activates Host Lysosomal Activity to Facilitate Lipid Breakdown from Worm to Mammalian Liver" for consideration as a Research Article by PLOS Biology.

Your manuscript has now been evaluated by the PLOS Biology editorial staff, as well as by an academic editor with relevant expertise, and I am writing to let you know that we would like to send your submission out for external peer review.

Once your full submission is complete, your paper will undergo a series of checks in preparation for peer review. After your manuscript has passed the checks it will be sent out for review. To provide the metadata for your submission, please Login to Editorial Manager (https://www.editorialmanager.com/pbiology) within two working days, i.e. by Aug 09 2025 11:59PM.

Kind regards,

Melissa

Melissa Vazquez Hernandez, Ph.D.

Associate Editor

PLOS Biology

---

## [Decision Letter · Decision Letter 1]

22 Sep 2025

Dear Bin,

Thank you for your patience while your manuscript "Microbial Tryptophan Metabolism Activates Host Lysosomal Activity to Facilitate Lipid Breakdown from Worm to Mammalian Liver" was peer-reviewed at PLOS Biology. Your manuscript has been evaluated by the PLOS Biology editors, an Academic Editor with relevant expertise, and by three independent reviewers.

As you will see in the reviewer reports, although the reviewers acknowledge the potential interest in your findings, they have also raised a substantial number of crucial concerns. Given our and the reviewer interest in your study, we would be open to inviting a comprehensive revision of the work; however, this would need to thoroughly address the reviewers' comments. Reviewer 1 mentions that specific metabolites responsible may still be undefined, and asks for clarification of which tissues show lysosomal phenotypes, more connection between worm findings and liver physiology, and tests of broader outcomes such as lifespan, stress resistance, and locomotion. Reviewer 2 finds the C. elegans experiments convincing but sees issues with the mouse studies. They ask for clarification of gavage frequency, consistency in cohort sizes, and note that antibiotic pretreatment only partially depleted the microbiota. They caution that the mutant strain may colonize differently than wild-type, which could confound results, and suggest quantifying colonization levels. Reviewer 3 argues that the mammalian data are not convincing enough to support broad claims. They caution against generalizing to “bacterial tryptophan metabolism” when only E. coli was tested, and call for more controls in worm studies. The reviewer stress that the mouse experiments lack critical controls (baseline groups, tryptophan-free conditions, careful quantification), making conclusions about conservation premature.

IMPORTANT: while we find that all suggestions will strengthen the work, we will leave to your discretion to identify the secondary metabolites. However, we want to highlight that you need to be very careful interpreting the mouse experiments, and addressing the experimental concerns and suggestions on mice experiments will be required for further consideration.

We appreciate that these requests represent a great deal of extra work, and we are willing to relax our standard revision time to allow you 6 months to revise your study. Please email us (plosbiology@plos.org) if you have any questions or concerns, or envision needing a (short) extension.

**IMPORTANT - SUBMITTING YOUR REVISION**

*Resubmission Checklist*

*Published Peer Review*

*PLOS Data Policy*

*Blot and Gel Data Policy*

Sincerely,

Melissa

Melissa Vazquez Hernandez, Ph.D.

Associate Editor

PLOS Biology

REVIEWERS' COMMENTS:

Reviewer #1:

The lysosomal LIPL-4-LBP-8 signaling pathway has previously been identified as a regulator of mitochondrial β-oxidation, lipid catabolism, and lifespan in C. elegans. In this pathway, LIPL-4 functions as a lysosomal acid lipase, while LBP-8 is a lipid chaperone protein that facilitates signal transduction from the lysosome to the nucleus. In this manuscript, the authors demonstrate that this pathway can be modulated by bacterial inputs. Specifically, C. elegans grown on wild-type K12 E. coli cultured in nutrient-rich LB medium show induction of LBP-8 and reduced lipid storage, effects that are suppressed in the tnaA mutant strain. Since tnaA encodes tryptophanase, which breaks down tryptophan into indole, pyruvate, and ammonium, its mutation likely increases tryptophan availability to C. elegans. Supporting this idea, the authors show that dietary tryptophan supplementation induces LBP-8 expression and nuclear translocation, while reducing lipid storage. These effects, however, were lost with dead E. coli, suggesting the involvement of secondary metabolites derived from bacterial tryptophan metabolism.

Next, the authors found that bacterial tryptophan inputs increase lysosomal acidification and degradation activities, and lysosomal functions including LIPL-4-mediated lysosomal lipolysis are required for tryptophan-induced lbp-8 up-regulation and lipid reduction. Consistent with prior findings that the LIPL-4-LBP-8 signaling pathway promotes mitochondrial beta-oxidation and mitoROS, the authors observed that bacterial tryptophan inputs enhance both processes. Moreover, the authors found that the effect of bacterial tryptophan inputs on boosting lysosomal functions and lipid catabolism is well conserved in hepatic cells, and high-fat-diet-fed mice supplemented with the tnaA E. coli mutant exhibited greater lipid accumulation in the liver compared to those supplemented with wild-type E. coli.

Overall, the manuscript is well written and presents solid data to support the authors' conclusions. The discovery that bacterial tryptophan inputs modulate lysosomal activity and activate the LIPL-4-LBP-8 signaling pathway is novel, and the evidence for conservation in mammalian systems is exciting. However, there are several major concerns that need to be addressed before this manuscript can be considered for publication at PLOS Biology.

Major points:

1. What is the secondary metabolite(s) derived from bacterial tryptophan metabolism that directly modulate LBP-8 levels and lipid storage? The authors showed that live E . coli is required for tryptophan supplementation to have an effect, suggesting secondary metabolites, rather than tryptophan itself, exert the regulatory effect on C. elegans. However, the specific metabolite(s) remain undefined. The authors should at minimum discuss potential candidates, e.g. possible hints from the KEIO screen.

2. In Figure 3, the authors presented result on lysosomal tubulation phenotype. This phenotype has been observed in the C. elegans hypodermis. The authors should clarify in which tissue the reported phenotype has been observed. If it is indeed in the hypodermis, the authors should examine lysosomal functions also in the intestine where lipid storage and lbp-8 expression have been examined.

3. In mice studies, lipid accumulation changes were observed in the liver, which is not the primary site to perceive bacterial tryptophan inputs. The authors should examine tryptophan levels in the circulation and provide discussion on how bacterial tryptophan inputs influence hepatic lipid metabolism. The authors should also consider examining lipid absorption in the intestine and circulating lipoprotein levels to strengthen the physiological relevance.

4. Given the role of the LIPL-4-LBP-8 signaling pathway in regulating lifespan and healthspan, the authors should assess whether bacterial tryptophan inputs affect worm lifespan, oxidative stress responses, and locomotion activity at later ages.

Minor points:

1. All the bar graphs should be changed to dot plot bar graphs to better represent data distribution.

2. Oleic acid instead of palmitic acid is commonly used to trigger lipid droplet formation in mammalian cells, as palmitic acid can cause lipotoxicity due to its poor incorporation into lipid droplets. The authors should utilize oleic acid in their hepatic cell culture experiments.

Reviewer #2:

The "Microbial Tryptophan Metabolism Activates Host Lysosomal Activity to Facilitate Lipid Breakdown from Worm to Mammalian Liver" by Zhang and colleagues proposes that bacterial tryptophan catabolism alters lysosomal activity and thereby fat accumulation. The paper is generally well written, the figures are clear, and the work is overall well presented. I am unqualified to assess the novelty of the lipid metabolism data and cannot comment on the interpretation of lysosomal architecture in Figure 3. Below I outline my impressions of the results.

The C. elegans data looks convincing. Since E. coli is the food source for C. elegans, it is plausible that different media and E. coli mutants lead to distinct metabolite profiles that then alter host metabolism. This could be further supported by metabolomic analyses focused on tryptophan metabolites from the different E. coli lawns, though as this point is implict it may be left out.

The in vivo data leave some open questions:

First, please clarify the frequency of gavage.

Second, the antibiotic pretreatment depleted only around 50% of gut bacteria, which seems mild. Fig. S6A reports this for fewer mice than those analyzed for hepatic lipid droplets in Fig. 6G. Please clarify the cohort inconsistency.

Third, it is likely that impaired tryptophan metabolism in the k/o mutant strain alters its ability to colonize the host. This would have downstream consequences on the range of effects triggered by Gram-negative proteobacterial colonization in the gut, including on the gut epithelium and immune system. Therefore, the authors should show that the different strains colonized the mice with similar efficiency, ideally through quantification of *E. coli* in the fecal samples of the mice analyzed here, or alternatively in new mice treated in the same way.

Last, the authors could consider analyzing portal vein blood to ascertain that gut microbial metabolism yielded different concentrations of tryptophan metabolites entering the liver.

Liver metabolism is not my area of expertise, but I am under the impression that tryptophan supplementation has been proposed in fatty liver disease. IAA as the product of tryptophan metabolism by diverse beneficial and pathogenic bacteria has been implicated in reducing hepatic lipogenesis, which the authors could discuss.

Overall, the experiments support the narrow claims of the authors. They acknowledge the critical limitations of their study, that the mechanism of tryptophan metabolite sensing is unknown. In addition, they do not resolve the specific (set of) metabolites.

Minor comments:

- L63: "engage in critical communication with bacteria": is unclear.

- L66: "the impact of bacterial metabolites on mitochondria is becoming increasingly in recent years": missing word?

- Fig 1C: metabolism misspelled

- L454: delete "This suggests a symbiotic relationship where gut microbes optimize lysosomal function to maintain metabolic homeostasis, contrasting with pathogenic subversion of lysosomal pathways." Symbiotic means "living together", and the implied mutualistic symbiosis is not demonstrated.

Reviewer #3 (François Leulier):

Reviewer report on "Microbial tryptophan metabolism activates host lysosomal activity to facilitate lipid breakdown from worm to mammalian liver" by Zhang et al. for PLOS Biology.

General comment:

Zhang et al. address both fundamental and clinically relevant questions that is the relationship between microbial metabolism and its consequences on host lipid metabolism. With in-depth mechanistic insights, the authors establish that E. coli tryptophan metabolism mediates an activation of C. elegans lysosomal activity through tryptophanase bacterial enzyme. They notably investigate and precisely characterize how lysosomal activity is stimulated upon microbial-mediated tryptophan metabolism. They interestingly translate their findings in cultured human hepatocytes, which broadens the discovery in another experimental setting. However, the data presented on mice fed high-fat diet does not allow to conclude because of both experimental settings and data analysis. Globally, the data presented and interpreted on C. elegans is robust, and convincing, with top-notch in vivo metabolic sensors. However, the translation in a mammalian system and the theoretically evolutionary conservation of the mechanism is still to be supported by more robust evidence.

Major comments:

1. Usage of "bacterial tryptophan metabolism"

Although the work provided by the authors about E. coli tryptophan metabolism is very robust, they only use one strain of E. coli with first a screen and then mostly the tnaA strain. The authors should then refrain from broadening their discovery to "bacterial tryptophan metabolism" in a general sense, or provide at least proof of concept that tryptophan metabolism of other bacterial species (eventually a commensal one, to increase the physiological relevance) also induces the same major changes in host lipid handling.

2. Use of appropriate controls in C. elegans study

When comparing the effect of microbial metabolism based on NGM versus LB medium worms and bacterial culture, the authors do not take into account: feeding behavior in front of dead/alive bacteria, is lbp-8 induction also dependant on feeding itself? Besides, they compare the physiological effect of E. coli cultured on NGM and LB, where the division dynamics are completely different: did the authors consider of bacterial cues coming from the different physiological states bacteria are in? They also need to include only LB control, in a richer medium it is very likely that the host metabolic baseline is changing independently of the bacterial presence.

3. Quantification of Oil Red O staining

The quantification of Oil Red O staining, as it is one of the primary sources of interpretation, should be more rigorous. Mean intensity is difficult to interpret as the blank can greatly vary between pictures (notably for mouse liver sections, but also visible on Figure 1B), and cannot be considered reliable. The authors should consider measuring the total area of lipid droplets, and/or the number/size of droplets observed. On the mice side, what is the NCD group that appears only once?

4. Design and interpretation of mice experiments

The absence of control group for bacterial gavage makes the interpretation impossible as there is no valuable baseline to compare to. Also, the absence of a group without tryptophan in drinking water prevents any interpretation as there is no evidence that the phenotype is not vastly tryptophan supplementation-mediated. The number of 6 mice per group is enough for a very strong phenotype, it is unsure whether this is the case in this study. For quantification purposes, the individual mouse should be considered as a variable during the analysis (Figure 6G), and the proper measurements used for statistical analysis should be indicated.

In vivo, there is no indication of tryptophan metabolism enhancement. Did the authors measure water intake in tryptophan supplemented water versus simple water? Did the authors make sure that bacterial gavage itself, and strain-specifically, did not induce changes in water intake? To ensure that it did not induce any preference leading to dissimilar tryptophan intake.

Interpretations and discussion about mouse experiment should be based on the scientific solidity of the experiments presented, the authors should give particular attention to this point. Especially, discussion and conclusions on hepatic steatosis and lipid clearance are not possible in this context: Oil Red O staining itself is not sufficient, other classical descriptive proxies such as adiposity, TAG levels in the liver and plasma were not reported.

Other comments:

- Can the other comment on the regionalization pattern of the reporter expression in Figure S2A, is it expected and can it be biologically meaningful?

- The preservation of a "youthful" lysosomal architecture is convincing, did the authors evaluate other proxies to quantify the "youthfulness" of lysosome in their experimental conditions.

- Although the authors mention that pH acidification is a proxy of lysosomal activity, do they have an idea of the effect size upon tryptophan supplementation? What is the optimal and physiological lysosomal acidification in conditions of individuals placed on LB-K12?

- How can the authors explain that in Figure 4C the Trp control line has Oil Red O intensity of 40, while in figure 4A the intensity if only 30 and considered a reduction compared to 40 in Mock-control?

- The Figure S6C should start at 0 and show absolute weights, as the authors mention that the animals are weighing the same. Why this panel contains only 4 points per condition as the authors mention using 6 in the Methods section? Interpretation on 4 liver weights, taking into account inter-individual variability prevents to conclude.

- The use of an antibiotic treatment before oral gavage of the bacterial solution should be justified.

- The authors should refrain from claiming "The majority of gut microbiota were eliminated", while the quantification clearly show a 3-fold decrease in bacterial 16S rRNA but no elimination (no idea about the other components of the microbiota). The microbiota is clearly highly disrupted, at least in numbers, but still present.

- Was the timing of gavage controlled? As the microbial load is changing during the day, as well as the whole community. The approximation of microbial load gavaged is not sufficient and cannot allow reproducibility. What exact concentration was given to each individual? And at what frequency was the solution given? The methods section should be clearer on this aspect, and the limitations acknowledged.

- Are AhR ligands only tryptophan-derived metabolites? The authors can detail on this.

- Is the association of such bacteria/microorganisms having this pathway and C. elegans happening in nature?

- Did the author choose to not investigate which tryptophan metabolites could be involved? Or did they not find anything compelling?

Technical comments:

- L136 error in "a reporter involved".

- Very small size of the text in Figure S5A complexifies the reading.

- L407 mention of results being in "hepatocytes", the authors should prefer "in Huh7 hepatocyte cell line".

- L212 mention of "bacteria TnaA is essential for a full transcriptional response to tryptophan" is misleading, the response is to tryptophan-metabolism derived compounds.

- Panel 6H does not provide supplementary information. If the authors wish to show images, then they should accompany them with rigorous quantifications.

Conclusion

Overall, this manuscript presents compelling and rigorous mechanistic work in C. elegans, which is very interesting and relevant for the readership of PLOS Biology. The strength of the study lies in the understanding of how E. coli tryptophan metabolism impacts lysosomal activity and lipid handling in the worm. However, several key issues must be addressed before the translational claims can be considered robust: (i) the scope of the findings should not be generalized to "bacterial tryptophan metabolism" without broader validation; (ii) the C. elegans experiments require appropriate controls to disentangle feeding behavior, bacterial physiological state, and medium effects; (iii) Oil Red O quantification needs to be more rigorous; and, (iv) most importantly, the mouse experiments lack essential controls, suffer from limited design and unclear description (including gavage procedures and water intake), and therefore do not allow conclusions about the evolutionary conservation of the mechanisms. Addressing these major points would considerably strengthen the manuscript and broaden its impact.

---

## [Decision Letter · Decision Letter 2]

4 Feb 2026

Dear Dr QI,

Thank you for your patience while we considered your revised manuscript "Microbial Tryptophan Metabolism Activates Host Lysosomal Activity to Facilitate Lipid Breakdown" for consideration as a Research Article at PLOS Biology. Your revised study has now been evaluated by the PLOS Biology editors, the Academic Editor and the original reviewers.

As you will see in the reports, all reviewers think recognize the good job done addressing their previous concerns, but Reviewers #2 and #3 still would like some aspects to be addressed. R2 acknowledges that the revised strategy using indole supplementation instead of microbial colonization is scientifically sound, but stresses that the manuscript repeatedly overstates the scope and evolutionary conservation of the findings by conflating bacterial tryptophan metabolism with indole treatment in mammals, and requests wording revisions and clearer limitation of claims. R3 indetified a statistical issue, as individual cells are treated as independent samples instead of using biological replicates (e.g., mouse or independent experiment) as the unit of analysis, which could artificially inflate significance and mislead conclusions.

In light of the reviews, which you will find at the end of this email, we are pleased to offer you the opportunity to address the remaining issues in a revision that we anticipate should not take you very long. We will then assess your revised manuscript and your response to the reviewers' comments with our Academic Editor aiming to avoid further rounds of peer-review.

**IMPORTANT - SUBMITTING YOUR REVISION**

*Resubmission Checklist*

*Published Peer Review*

*PLOS Data Policy*

*Blot and Gel Data Policy*

Sincerely,

Melissa

Melissa Vazquez Hernandez, Ph.D.

Associate Editor

PLOS Biology

REVIEWERS' COMMENTS:

Reviewer #1:

The authors have fully addressed my comments, and I highly recommend the manuscript to be published at PLOS Biology.

Reviewer #2:

The authors have addressed prior concerns by replacing problematic mouse colonization experiments with indole supplementation studies in hepatocytes and mice. This revision strategy avoids earlier methodological issues and is scientifically valid, though it narrows the claim because the revised data show effects of indole administration in mammals, not of bacterial tryptophan metabolism.

Along this line, the revised manuscript overstates the scope of these findings in several points, requiring correction:

- L21: States "bacterial tryptophan metabolism activates lysosomal activity" in the context of the phrase "evolutionary conserved", implying that findings extend to mammals, but the mammalian data demonstrate effects of indole supplementation, not bacterial metabolism per se.

- L369-440: "Bacterial tryptophan metabolism enhances lysosomal function which promotes lipid breakdown in hepatocyte cell line". The experiments in this section use indole treatment or bacterial supernatants, not colonization or active metabolism.

- L504 Discussion: Refers to "bacterial indole" regulating host lipid homeostasis when the mammalian experiments administered purified indole.

- Discussion L492 ff and Figure 8 and legend: The way this model is used in the narrative overreaches the data. The model in Figure 8 highlights tnaA, and the phrasing in the discussion "Our study uncovers a conserved host-microbe metabolic axis" extends this across host species, which was not shown.

These statements should be revised to accurately reflect that the C. elegans data demonstrate bacterial tryptophan metabolism effects, but the mammalian data demonstrate effects of indole. The author may consider a limitations section softening the not fully supported claims surrounding a conserved axis as a whole.

The original claim of a "symbiotic relationship where gut microbes optimize lysosomal function" was removed. However, similar language appears again at L515, now framing this as "This symbiotic relationship between indole-producing bacteria and C. elegans supports the physiological relevance of our findings that bacterial indole regulates host lipid metabolism through lysosomal activation." Several issues:

1) The authors report that live E. coli cultured on tryptophan-supplemented media induce lbp-8 more strongly than heat-killed bacteria (L140)m, but it is unclear to me whether the heat killed bacteria were exposed to tryptophan prior to inactivation. Would this not mean that only the live bacteria would have had the opportunity to convert tryptophan into indole? This should be clarified to understand if this assay can truly isolate the role of bacterial viability or metabolism.

2) While the "symbiotic relationship" phrasing is technically limited to C. elegans, the discussion that follows (comparing indole effects to pathogen subversion of lysosomal pathways) implicitly elevates this interaction to a conserved host-microbe dynamic. However, the worm model involves ingestion and digestion of bacteria, and the host responds to metabolites. In mammals, bacterial metabolism was not tested. Without evidence that similar interactions occur in a live microbial context in mammals, the analogy to pathogen-host strategies creates an impression of evolutionary conservation that overreaches the data.

Minor issues:

Figure 7F/G: this seems to show pooled data across three mice. Given the very small n of mice, it is possible that the slightly elevated levels are driven by single animals in either group. The authors should show the data from each mouse (i.e. six bars, three from each arm). Furthermore, they may consider mixed effects models for repeated measurements from the same mice to compare groups, e.g. intensity ~ 1 + IndoleTreatment + (1 | mouseID)

L211 : "…as the specific secondary" should be "as a specific…"

Reviewer #3 (François Leulier):

The authors provide extensive and thoughtful responses to both the major and minor concerns raised in the initial review. Most of the issues have been addressed appropriately, and the claims now presented appear scientifically well supported. The narrative is now coherent and complete, with adequate controls and mechanistic insights in the C. elegans data, as well as carefully interpreted results from hepatocyte experiments (both primary cells and cell lines). I understand the authors' decision to remove the in vivo mouse experiments for both technical and conceptual reasons, and the claims have been adjusted accordingly. Overall, the manuscript has become much clearer, with a simplified experimental design that still maintains strong biological relevance.

The only remaining major concern relates to the statistical analysis. The authors need to carefully consider the biological unit relevant to their conclusions. For instance, in the case of mouse-derived hepatocytes, the biological unit of interest is not the individual cell but the population of cells originating from a single mouse. Applying standard statistical analyses based on large numbers of cells (e.g., n = 491 and n = 571 cells in Fig. 7F) is inappropriate and artificially inflates statistical power by increasing the sample size without increasing the number of independent biological replicates. Individual cells are not independent biological entities in this context; the mouse is.

Moreover, the authors state in the Methods that "all experiments were performed independently at least three times with similar results." Therefore, statistical analyses should be performed across biological replicates, using the independent experiment as the unit of analysis rather than individual cells. An incorrect statistical approach would lead to exaggerated significance levels and potentially misleading conclusions.

---

## [Editor Report · Decision Letter 3]

11 Feb 2026

Dear Bin,

Thank you for your patience while we considered your revised manuscript "Microbial Tryptophan Metabolism Activates Host Lysosomal Activity to Facilitate Lipid Breakdown" for publication as a Research Article at PLOS Biology. This revised version of your manuscript has been evaluated by the PLOS Biology editors, and the Academic Editor.

Based on our Academic Editor's assessment of your revision, we are likely to accept this manuscript for publication, provided you satisfactorily address the remaining editorial points. If you require more time for revision giving the upcoming chinese festivities, please let me know.

1) Please add weblink of the funding agencies in the Financial Disclosure statement in the manuscript details during submission and in the main text.

2) The Ethics statement needs to be a separate, independent (and the first) subheading in the Material & Methods section. It must include the full name of the IACUC/ethics committee that reviewed and approved the animal care and use, as well as the protocol/permit/project license number. https://journals.plos.org/plosbiology/s/ethical-publishing-practice

3) Please cite the location of the data clearly in all relevant main and supplementary Figure legends, e.g. “The data underlying this Figure can be found in S1 Data” or “The data underlying this Figure can be found in https://doi.org/10.5281/zenodo.XXXXX”

4) Supplementary files (e.g., excel). Please ensure that all data files are uploaded as 'Supporting Information' and are invariably referred to (in the manuscript, figure legends, and the Description field when uploading your files) using the following format verbatim: S1 Data, S2 Data, etc. Multiple panels of a single or even several figures can be included as multiple sheets in one excel file that is saved using exactly the following convention: S1_Data.xlsx (using an underscore).

5) Please ensure that your Data Statement in the submission system accurately describes where your data can be found and is in final format, as it will be published as written there

6) Per journal policy, if you have generated any custom code during the course of this investigation, please make it available without restrictions. Please ensure that the code is sufficiently well documented and reusable, and that your Data Statement in the Editorial Manager submission system accurately describes where your code can be found. More information on our Code Policy, what and how to share can be found here: https://journals.plos.org/plosbiology/s/code-availability

We expect to receive your revised manuscript within two weeks.

*Published Peer Review History*

*Press*

Sincerely,

Melissa

Melissa Vazquez Hernandez, Ph.D.

Associate Editor

PLOS Biology

---

## [Editor Report · Decision Letter 4]

16 Feb 2026

Dear Bin,

Thank you for the submission of your revised Research Article "Microbial Tryptophan Metabolism Activates Host Lysosomal Activity to Facilitate Lipid Breakdown" for publication in PLOS Biology. On behalf of my colleagues and the Academic Editor, Ken Cadwell, I am pleased to say that we can in principle accept your manuscript for publication, provided you address any remaining formatting and reporting issues. These will be detailed in an email you should receive within 2-3 business days from our colleagues in the journal operations team; no action is required from you until then. Please note that we will not be able to formally accept your manuscript and schedule it for publication until you have completed any requested changes.

PRESS

Sincerely,

Melissa

Melissa Vazquez Hernandez, Ph.D., Ph.D.

Associate Editor

PLOS Biology
